# The reactive pyruvate metabolite dimethylglyoxal mediates neurological consequences of diabetes

Sina Rhein[1,2,13], Riccardo Costalunga[1,2,3,13], Julica Inderhees [1,2,3,13], Tammo Gürtzgen[1], Teresa Christina Faupel [1], Zaib Shaheryar [1], Adriana Arrulo Pereira[1], Alaa Othman [3,12], Kimberly Begemann[1], Sonja Binder[1], Ines Stölting[1], Valentina Dorta[4], Peter P. Nawroth[5], Thomas Fleming [5,6], Konrad Oexle[7,8], Vincent Prevot [9], Ruben Nogueiras [4], Svenja Meyhöfer[6,10,11], Sebastian M. Meyhöfer[6,10] & Markus Schwaninger [1,2] ✉

Complications of diabetes are often attributed to glucose and reactive dicarbonyl metabolites derived from glycolysis or gluconeogenesis, such as methylglyoxal. However, in the CNS, neurons and endothelial cells use lactate as energy source in addition to glucose, which does not lead to the formation of methylglyoxal and has previously been considered a safer route of energy consumption than glycolysis. Nevertheless, neurons and endothelial cells are hotspots for the cellular pathology underlying neurological complications in diabetes, suggesting a cause that is distinct from other diabetes complications and independent of methylglyoxal. Here, we show that in clinical and experimental diabetes plasma concentrations of dimethylglyoxal are increased. In a mouse model of diabetes, ilvb acetolactate-synthase-like (ILVBL, HACL2) is the enzyme involved in formation of increased amounts of dimethylglyoxal from lactate-derived pyruvate. Dimethylglyoxal reacts with lysine residues, forms Nε−3-hydroxy-2-butanonelysine (HBL) as an adduct, induces oxidative stress more strongly than other dicarbonyls, causes blood-brain barrier disruption, and can mimic mild cognitive impairment in experimental diabetes. These data suggest dimethylglyoxal formation as a pathway leading to neurological complications in diabetes that is distinct from other complications. Importantly, dimethylglyoxal formation can be reduced using genetic, pharmacological and dietary interventions, offering new strategies for preventing CNS dysfunction in diabetes.

Diabetes complications place a great burden on patients and cause a lot of suffering. Some progress has been made to prevent and treat the classical micro- and macrovascular sequelae of diabetes. However, the situation is still unsatisfactory for CNS involvement. So far, there is no medication to prevent or improve cognitive impairment in patients with diabetes[1]. In addition, hyperglycemia in acute stroke leads to a worsening of the clinical condition and increased mortality. However, normalization of high glucose levels with insulin does not improve the outcome of stroke patients with hyperglycemia[2]. These therapeutic deficits challenge the current view, derived from studies looking mostly at peripheral vascular pathology, that hyperglycemia- and glycolysis- (or gluconeogenesis)-dependent reactions drive diabetic

CNS complications or the detrimental effects of acute hyperglycemia in stroke[3–5]. Instead, we hypothesize that specific features in the development of CNS complications could be due to differences in the energy supply of peripheral and neural cells.

In peripheral tissues, there is good evidence that reducing sugars, including extracellular glucose and its reactive metabolites, mediate diabetic complications[3,4,6]. These metabolites encompass the α-dicarbonyl compounds methylglyoxal, glyoxal, and 3-deoxyglucosone. They are known to be generated by non-enzymatic pathways during gluconeogenesis and glycolysis[7,8]. α-Dicarbonyls react with lysine, arginine, and cysteine residues and thereby change the function of enzymes and other proteins[9,10]. In addition, the modified proteins, referred to as advanced glycation end products (AGEs), induce oxidative stress and promote inflammation, both of which contribute to tissue injury[11–14]. The energy metabolism of neurons and brain endothelial cells differs from other cell types in one aspect important for this study. While glia and many other cells use glucose and heavily depend on glycolysis resulting in lactate release[8,15], neurons and to some degree also CNS endothelial cells take up lactate with the help of monocarboxylate transporters 1–4 and survive on this glucose derivative[16–19]. Because glycolysis occurs in other cells, it is expected that lactate-consuming cells produce less 3-deoxyglucosone, glyoxal, and methylglyoxal than cells that survive via glycolysis[8,20]. If this were true, neurons should be protected from diabetic complications. On the contrary, neurons and brain endothelial cells are primary sites for the development of diabetic complications[1]. This makes it likely that mechanisms other than those described for peripheral tissue are operative in the CNS. Therefore, we searched for pathways generating reactive metabolites that induce neuronal carbonylation stress independent from glycolysis and elevated extracellular glucose. Using metabolic tracing, we found that dimethylglyoxal, an α-dicarbonyl previously not studied in the context of diabetic complications, is elevated under hyperglycemic and hypoxic conditions. Dimethylglyoxal turned out to be an interesting candidate, since its generation was only remotely related to glycolysis, but closely linked to the uptake and metabolism of pyruvate. Its accumulation is under enzymatic control and it can cause key cognitive complications associated with diabetes. Importantly, dietary, genetic, and pharmacological interventions were able to reduce dimethylglyoxal production, potentially offering innovative strategies to prevent diabetic CNS complications.

## Results

### Elevated levels of the α-dicarbonyl dimethylglyoxal in diabetes

To measure α-dicarbonyls in biological samples, we performed a derivatizing reaction with o-phenylenediamine and detected the resulting quinoxaline products with liquid chromatography-mass spectrometry (LC-MS) as described previously[21–23]. As internal standard, we initially used the commercially available compound 2,3-dimethylquinoxaline (2,3-DMQ). However, when we established the method with plasma from mice, one week after inducing diabetes with streptozotocin (STZ, Fig. 1a, b), we observed higher 2,3-DMQ levels in the STZ group (Fig. 1c). This suggested that STZ raised blood concentrations of either 2,3-DMQ itself or of dimethylglyoxal, the α-dicarbonyl that generates 2,3-DMQ upon o-phenylenediamine derivatization (Fig. 1d). To distinguish between these options, we replaced the derivatizing agent and the internal standard by deuterated o-phenylenediamine (d8-DB) and deuterated methylglyoxal (d4-methylglyoxal), respectively. The new analyses revealed that the derivatization product of dimethylglyoxal and d8-DB was increased (Supplementary Fig. 1a), demonstrating that diabetic mice have elevated plasma dimethylglyoxal concentrations. One week after treating mice with STZ, a model for type-1 diabetes[24], circulating dimethylglyoxal concentrations were increased 4.8-fold, while methylglyoxal was not elevated. Dimethylglyoxal concentrations in plasma were still

elevated after STZ-induced chronic hyperglycemia for 14–15 weeks (Fig. 1e; Supplementary Fig. 1b, c). To assess potential differences between periphery and CNS, α-dicarbonyls were also measured in whole-brain extracts, where the proportional increase in dimethylglyoxal levels was at least as high as that of 3-deoxyglucosone, glyoxal, and methylglyoxal (Fig. 1f and Supplementary Fig. 1d). Dimethylglyoxal plasma concentrations were also increased in leptin receptor-deficient *db/db* mice, an established type-2 diabetes mouse model, comparable to 3-deoxyglucosone and methylglyoxal (Fig. 1g and Supplementary Fig. 1e, f)[25]. As *db/db* mice are obese and hyperglycemic, we also investigated mice with high-fat diet-induced obesity, a mouse model of impaired glucose tolerance[26] (Supplementary Fig. 1g). After a 12-week high-fat diet, dimethylglyoxal plasma concentrations remained unaltered, indicating that overt type-2 diabetes, rather than impaired glucose tolerance or obesity, increases dimethylglyoxal plasma concentrations.

Dimethylglyoxal has been detected in human blood[27] but has not yet been linked to diabetes; therefore, we tested whether dimethylglyoxal is a clinically relevant metabolite in patients with diabetes. Demographic parameters are listed in Supplementary Table 4. As expected, glucose, 3-deoxyglucosone, glyoxal, and methylglyoxal levels were higher in patients with diabetes (Fig. 2a). Consistent with our mouse data, patients with diabetes also showed a marked increase in serum dimethylglyoxal (Fig. 2a). However, in contrast to 3-deoxyglucosone but similar to glyoxal and methylglyoxal, dimethylglyoxal concentrations did not correlate with glucose serum levels (Supplementary Fig. 1h). Nevertheless, an analysis of receiver operating characteristics demonstrated that dimethylglyoxal had a remarkable diagnostic utility in classifying patients as diabetic, which tends to be better than that of glyoxal or methylglyoxal (Supplementary Fig. 1i). Subgroup analysis demonstrated that glyoxal and methylglyoxal were elevated only in patients with type-2 diabetes. In contrast, for dimethylglyoxal we found higher concentrations in type-1 and type-2 diabetes, similar to 3-deoxyglucosone (Fig. 2b). To approach the question of how to lower dimethylglyoxal levels, we were interested in the effects of the main antidiabetic drugs, insulin, metformin, or the combination of both. While the three treatment regimens did not differ in controlling hyperglycemia, the combination of metformin and insulin seemed superior in reducing dimethylglyoxal concentrations compared with insulin alone; the effect on 3-deoxyglucosone and glyoxal was similar, whereas there was no effect on methylglyoxal (Fig. 2c). Overall, these data indicate, that in human and experimental diabetes dimethylglyoxal increases significantly and often more than glycolysis-dependent dicarbonyls, with a marked increase in the brain. Furthermore, the rise in dimethylglyoxal does not directly depend on hyperglycemia.

Very little is known about the regulation and function of endogenous dimethylglyoxal. To better understand mechanisms affecting dimethylglyoxal plasma levels, we fed non-diabetic and diabetic mice a ketogenic diet with low carbohydrate and protein content (Fig. 3a), which is known to reduce glucose utilization and to lower blood glucose concentrations in diabetes[28,29]. In non-diabetic mice, the ketogenic diet did not change glucose concentrations (Fig. 3b) but markedly lowered dimethylglyoxal concentrations in plasma (Fig. 3d), likely reflecting the reduced glucose utilization during ketosis[28]. While ketogenic diet had no effect on the STZ-induced drop of insulin levels (Fig. 3c), it did normalize blood glucose concentrations in the STZ-treated mice at day 56, as reported previously[29] (Fig. 3b). The ketogenic diet also normalized glucose levels in urine at day 56 and the daily intake of food and water in STZ-treated mice (Supplementary Fig. 1j, k). Importantly, the diet efficiently reduced dimethylglyoxal levels in the plasma of diabetic STZ-treated mice (Fig. 3d), in parallel to lowering glucose concentrations, thus highlighting the association between diabetes and dimethylglyoxal production. In contrast, glyoxal and methylglyoxal were not affected by the ketogenic diet (Fig. 3d).

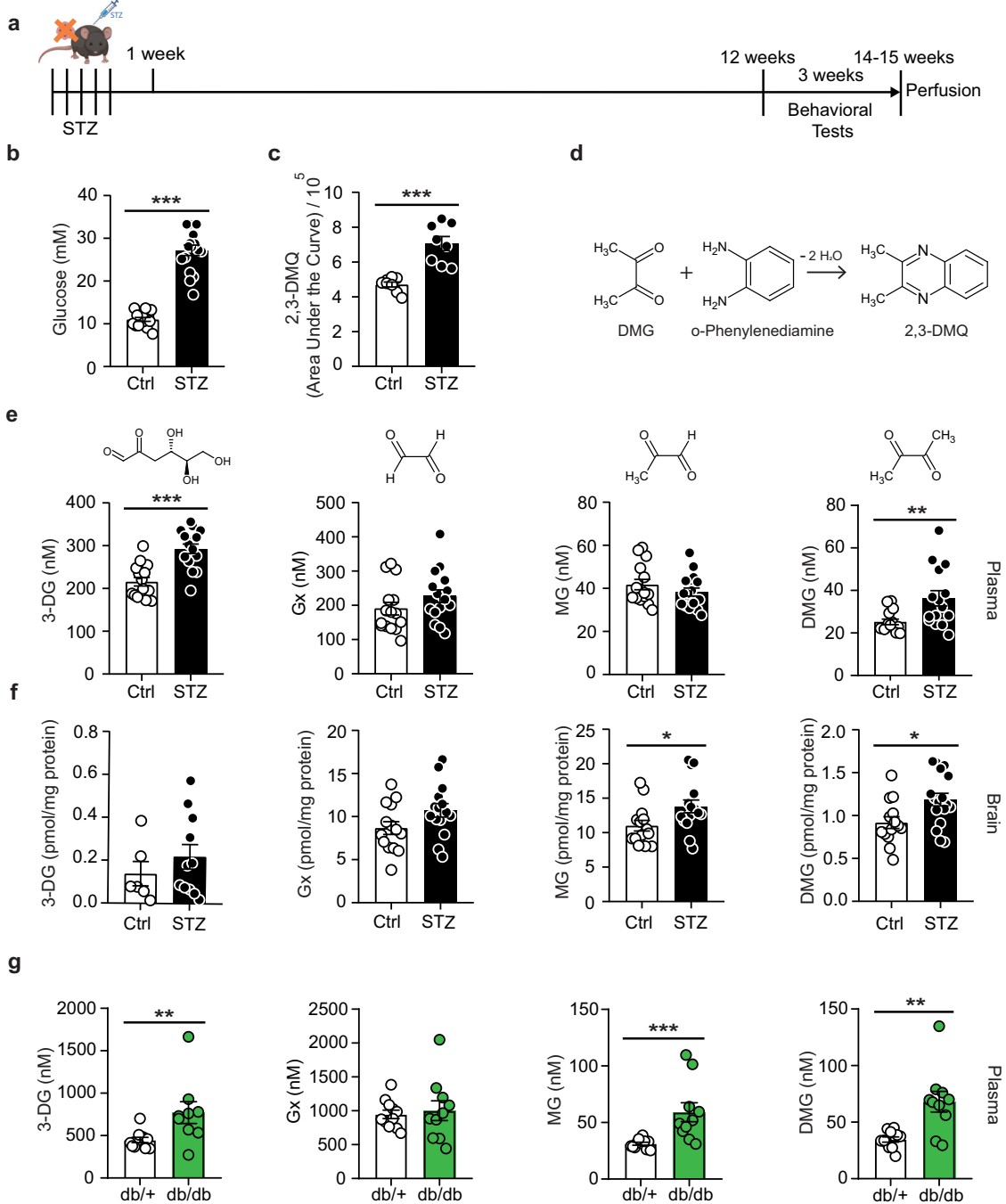

**Fig. 1 | Experimental diabetes increases dimethylglyoxal concentrations.**
**a** Timeline of streptozotocin (STZ)-induced diabetes, behavioral tests, and sample collection in mice. Created with BioRender.com, released under a Creative Commons Attribution-NonCommercial-NoDerivs 4.0 International license.
**b** Blood glucose levels 14 weeks after STZ ($n = 17$) and control ($n = 15$) injections.
**c** Levels of 2,3-dimethylquinoxaline (2,3-DMQ) in mouse plasma 1 week after STZ injections ($n = 8$ per group). **d** 2,3-DMQ is formed by the reaction of dimethylglyoxal (DMG) with o-phenylenediamine. **e** Concentrations of 3-deoxyglucosone (3-DG), glyoxal (Gx), methylglyoxal (MG), and DMG in mouse plasma 14 weeks after vehicle ($n = 15$) or STZ ($n = 15$-17) injections. **f** Concentrations of 3-DG, Gx,

MG, and DMG in mouse brain 14 weeks after vehicle ($n = 14$ except $n = 6$ for 3-DG) or STZ ($n = 15$-17 except $n = 11$ for 3-DG) injections. **g** Plasma concentrations of 3-DG, Gx, MG, and DMG in leptin receptor-deficient (*db/db*; $n = 9$–10) mice and heterozygous littermates (*db/+*; $n = 11$) at an age of 10–11 weeks. For normalization, deuterated methylglyoxal (d4-MG) was used as internal standard (**e**–**g**). Values are means ± SEM. Statistical comparison by two-tailed Mann–Whitney *U* test (**b**, **c**, **e** (DMG) and **g**) or two-tailed unpaired *t* test (**e**, **f**). *$p < 0.05$; **$p < 0.01$; ***$p < 0.001$. Detailed information on the test statistics is provided in Supplementary Table 5. Source data are provided as a Source Data file.

## Dimethylglyoxal is derived from glucose and pyruvate

Elevated concentrations in diabetes raised the possibility that dimethylglyoxal is a product of glucose metabolism. To test this idea, we treated brain endothelial bEnd.3 cells with native [$^{12}$C]- or uniformly labeled [$^{13}$C$_6$]-glucose for 24 h. By detecting a mass increase in α-

dicarbonyls secreted to the medium using high-resolution MS, we were able to trace carbon atoms derived from [$^{13}$C$_6$]-glucose. Dimethylglyoxal incorporated $^{13}$C in two or four and only to a low degree in three C-atoms, suggesting that dimethylglyoxal is produced by the reaction of two C-C blocks derived from glucose (Fig. 4a). The

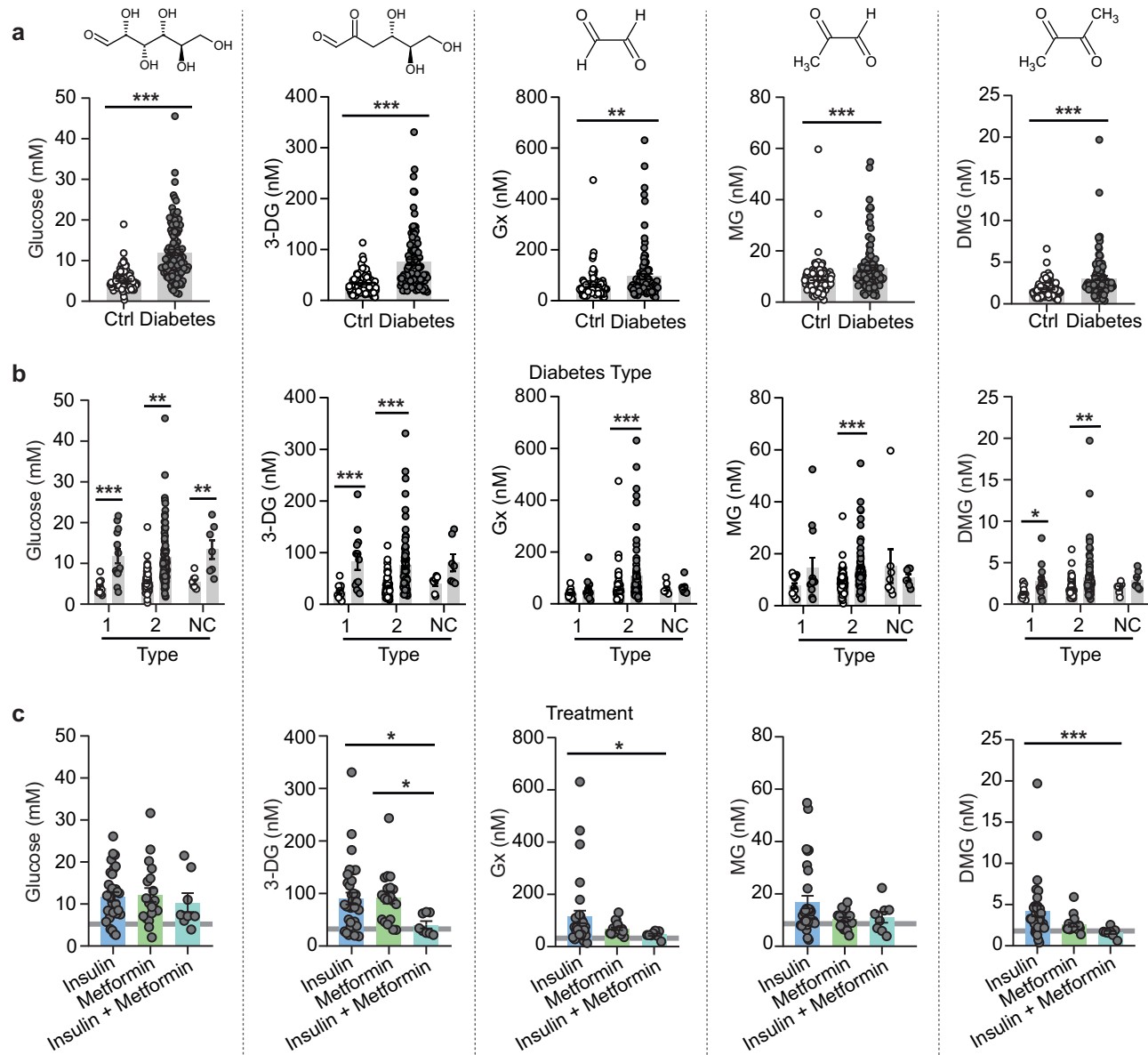

**Fig. 2 | Dimethylglyoxal is elevated in patients with diabetes. a** Comparison of glucose, 3-deoxyglucosone (3-DG), glyoxal (Gx), methylglyoxal (MG), and dimethylglyoxal (DMG) concentrations in serum of controls ($n = 100$) and patients with diabetes ($n = 98$–100). **b** Alpha-dicarbonyl concentrations in serum of patients with diabetes type-1 ($n = 13$), type-2 ($n = 75$–77), or non-classified (NC, $n = 7$) diabetes as well as age- and sex-matched control subjects without diabetes ($n = 14/77/8$). **c** Effect of antidiabetic treatment on serum concentrations of DMG and other α-dicarbonyls. Patients with diabetes were grouped according to their treatment ($n = 32$–33 (insulin), $n = 19$ (metformin), $n = 8$ (insulin+metformin)). The gray line indicates concentrations in control patients ($n = 100$). Values are means ± SEM. Statistical comparison by two-tailed Mann–Whitney $U$ test (**a**), Scheirer-Ray-Hare test followed by two-tailed Mann–Whitney $U$ post hoc tests and Bonferroni-Holm correction (**b**), or Kruskal–Wallis followed by Dunn's post hoc tests (**c**). *$p < 0.05$; **$p < 0.01$; ***$p < 0.001$. Detailed information on the test statistics is provided in Supplementary Table 5. Detailed information on patient characteristics is provided in Supplementary Table 4. Source data are provided as a Source Data file.

concentrations of labeled α-dicarbonyls in cell extracts were below the detection limit of the applied method.

As hypoxia increases anaerobic glycolysis and promotes methylglyoxal production[7], we wondered whether it would also affect dimethylglyoxal production. bEnd.3 cells were treated with [$^{13}C_6$]-glucose at atmospheric or low (0.1–0.5%) $O_2$ concentrations for 6 h. Every 30 min, we measured $^{13}C$-labeled α-dicarbonyls in the medium (Fig. 4b, c and Supplementary Fig. 2a, b). As expected, hypoxia increased the production of [$^{13}C_3$]-methylglyoxal and also had a pronounced effect on uniformly labeled dimethylglyoxal. To test whether hypoxia/ischemia would enhance dimethylglyoxal production from glucose in vivo, we used a mouse model of acute ischemic stroke under hyperglycemia, a common constellation in

stroke patients[30]. To this end, we injected glucose (50 mg) or vehicle in mice and induced strokes by occluding the middle cerebral artery (MCAO, Fig. 4d). In this model, glucose administration increased blood glucose levels from 6 mM to 10.7 mM[14]. Dimethylglyoxal levels were elevated in plasma 50 min after treating mice with vehicle, but were even higher after glucose administration (Fig. 4e). During acute hyperglycemia, dimethylglyoxal levels were higher in the ischemic ipsilateral hemisphere of the brain than in the contralateral side (Fig. 4f), indicating that hypoxia might promote dimethylglyoxal production from glucose in vivo as well.

In order to test whether glycolysis is linked to the production of dimethylglyoxal, cells were treated with 2-deoxyglucose, an inhibitor of hexokinase and phosphoglucose isomerase, in addition to hypoxia.

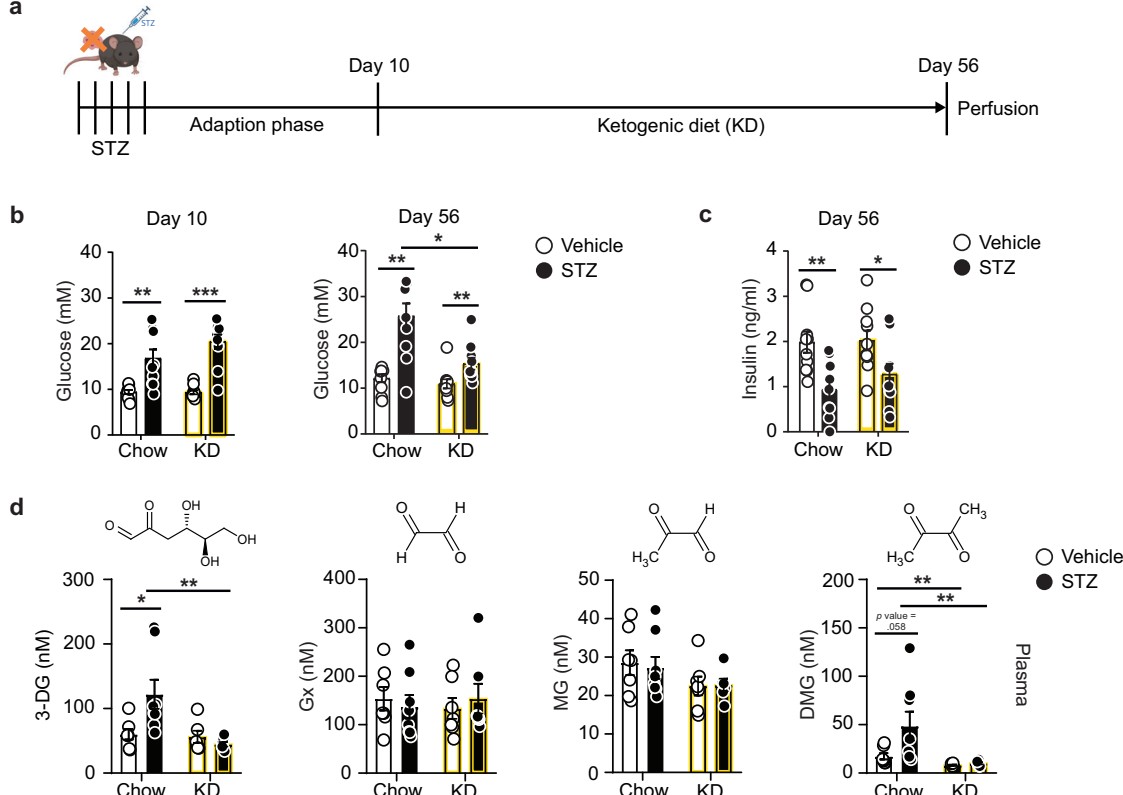

**Fig. 3 | Ketogenic diet normalizes dimethylglyoxal concentrations. a** Timeline of streptozotocin (STZ)-induced diabetes, ketogenic diet (KD), and sample collection in mice. Created with BioRender.com, released under a Creative Commons Attribution-NonCommercial-NoDerivs 4.0 International license. **b** Glucose concentrations in blood 10 and 56 days after vehicle or STZ injections ($n = 10$). **c** Insulin concentrations in plasma 56 days after vehicle ($n = 10$) or STZ ($n = 9$–10) injections. **d** Concentrations of 3-deoxyglucosone (3-DG), glyoxal (Gx), methylglyoxal (MG),

and dimethylglyoxal (DMG) in mouse plasma 56 days after vehicle ($n = 6$–7) or STZ ($n = 6$–8) injections. Values are means ± SEM. Statistical comparison by Scheirer-Ray-Hare test followed by two-tailed Mann–Whitney $U$ post hoc tests and Bonferroni-Holm correction (**b**, **d** (3-DG and MG)) or 2-way ANOVA followed by Sidak's post hoc tests (**c**, **d** (Gx and MG)). *$p < 0.05$; **$p < 0.01$; ***$p < 0.001$. Detailed information on the test statistics is provided in Supplementary Table 5. Source data are provided as a Source Data file.

As 3-deoxyglucosone and glyoxal are formed independent of glycolysis[31,32], [$^{13}C_6$]-3-deoxyglucosone and [$^{13}C_2$]-glyoxal were not affected by the treatment (Supplementary Fig. 2c, d). However, 2-deoxyglucose completely prevented the formation of [$^{13}C$]-methylglyoxal after 5 h (Supplementary Fig. 2e, f). In line with the observation that the indirect influence of 2-deoxyglucose on downstream effects in glycolysis decreases with their metabolic distance from the upstream steps that it directly inhibits[33], 2-deoxyglucose resulted in a moderate decrease of [$^{13}C_3$]-pyruvate (Supplementary Fig. 2i). In contrast, the effect of 2-deoxyglucose on $^{13}C$-labeled dimethylglyoxal was slight and only fully labeled [$^{13}C_4$]-dimethylglyoxal was lowered (Supplementary Fig. 2g, h), arguing that dimethylglyoxal is a distant glucose metabolite. In line with this, the delay in accumulation of [$^{13}C$]-labeled methylglyoxal, pyruvate, and dimethylglyoxal reflects an increasing number of metabolic steps downstream of [$^{13}C_6$]-glucose (Supplementary Fig. 2j–l).

In prokaryotes and yeast, dimethylglyoxal is formed during pyruvate metabolism[34,35]. Therefore, we determined whether pyruvate is also a precursor of dimethylglyoxal in mammalian cells. We treated bEnd.3 cells with [$^{13}C_1$-1]-, [$^{13}C_1$-2]-, [$^{13}C_1$-3]- or [$^{13}C_3$-1,2,3]-pyruvate for 24 h and compared the enrichment of [$^{13}C$]-dimethylglyoxal to control cells that were treated with [$^{12}C_3$]-pyruvate. [$^{13}C_1$-2], [$^{13}C_1$-3] and [$^{13}C_3$-1,2,3]-pyruvate, but not [$^{13}C_1$-1]-pyruvate enriched half-labeled [$^{13}C_2$]-dimethylglyoxal (Fig. 5a), indicating that C-1 of pyruvate is not incorporated in the dimethylglyoxal structure. Fully labeled [$^{13}C_4$]-dimethylglyoxal was only found if cells had been treated with fully labeled [$^{13}C_3$-1,2,3]-pyruvate (Fig. 5a), in line with the concept that

dimethylglyoxal is a reaction product of two pyruvate molecules after decarboxylation of C-1 (Fig. 5b). Since no dimethylglyoxal was formed in parallel cell-free wells, chemical degradation of pyruvate could be excluded as a source of dimethylglyoxal production in these experiments (Fig. 5c). The finding that dimethylglyoxal is a pyruvate metabolite raised the question of whether dimethylglyoxal concentrations are higher in diabetes due to higher lactate or pyruvate concentrations. Increased plasma lactate concentrations in clinical diabetes are reported in the literature, which supports this idea[36]. Besides its function as energy substrate in the brain, lactate is a primary substrate of the tricarboxylic acid cycle in peripheral tissues[15] suggesting lactate/pyruvate metabolism as a peripheral source of dimethylglyoxal. However, in the plasma of STZ-treated mice we found slightly lower pyruvate concentrations and an increased ratio of lactate to pyruvate (Supplementary Fig. 1c), probably reflecting a lower NAD$^+$/NADH ratio[37]. Since slightly lower pyruvate levels in plasma obviously cannot explain the increased dimethylglyoxal concentrations, other compartments than plasma might be more relevant for dimethylglyoxal production.

The reaction of two pyruvate molecules can be catalyzed by pyruvate dehydrogenase, the E1 subunit of the pyruvate dehydrogenase complex (PDC). In addition to synthesizing acetyl-CoA, PDC has been reported to produce acetate, acetoin, or acetaldehyde[38–42]. Acetoin is readily converted into its oxidized form, dimethylglyoxal[43]. To investigate the involvement of PDC in dimethylglyoxal production, we activated PDC with dichloroacetate (DCA) in brain endothelial cells. DCA lowered the concentrations of [$^{13}C_3$]-pyruvate derived from [$^{13}C_6$]-

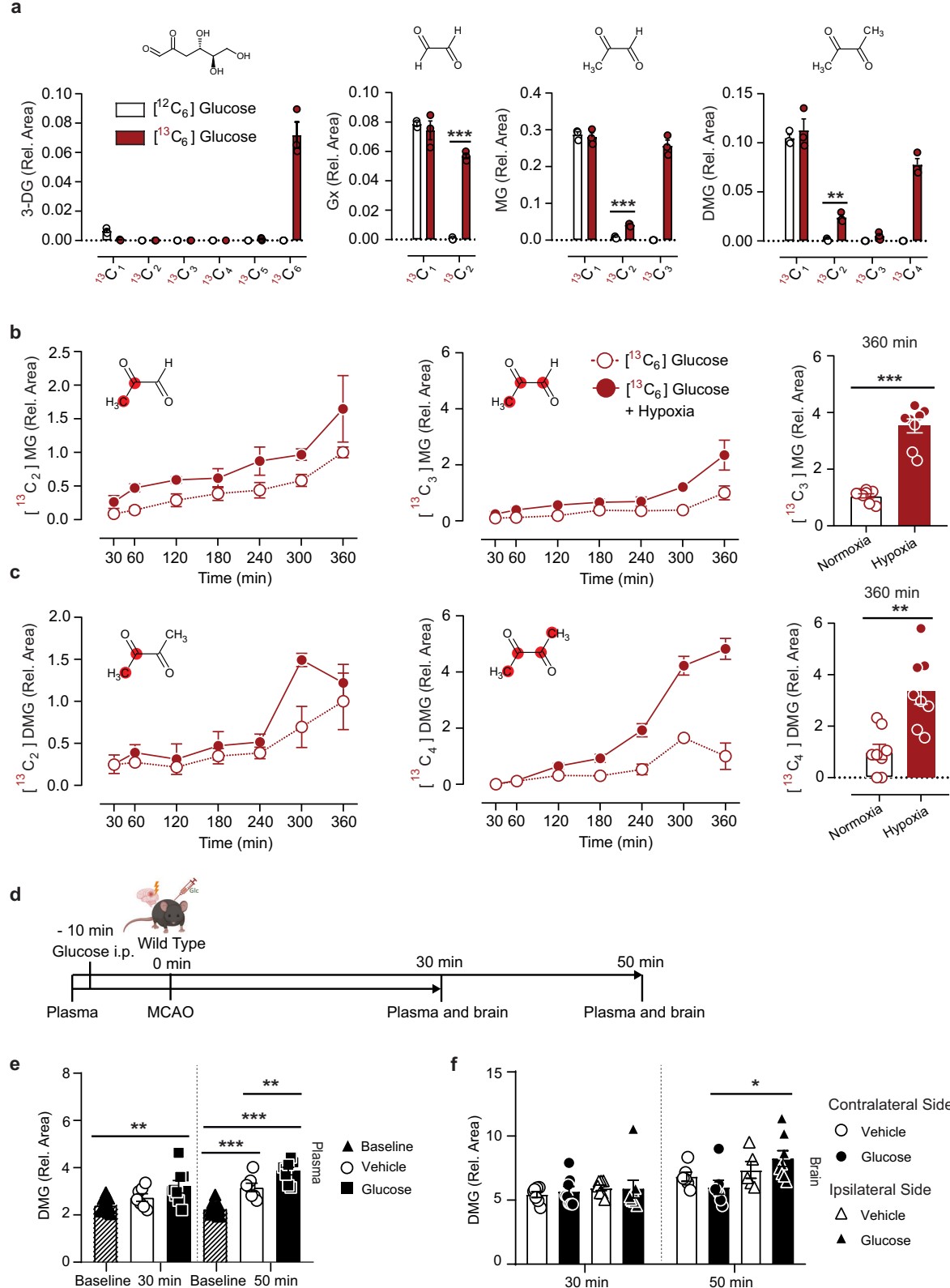

glucose, in accordance with a faster turnover of pyruvate by activated PDC, but had no significant effect on [$^{13}C_2$]- or [$^{13}C_4$]-dimethylglyoxal levels (Supplementary Fig. 2m, n). Inhibiting PDC by hypoxia[44,45], in contrast, elevated [$^{13}C_3$]-pyruvate and [$^{13}C_4$]-dimethylglyoxal (Fig. 4c, Supplementary Fig. 2m, n), arguing that PDC does not form dimethylglyoxal under the conditions tested here.

## ILVBL is involved in dimethylglyoxal formation in the brain

In prokaryotes, plants, and fungi, dimethylglyoxal is a byproduct of branched-chain amino acid synthesis that starts with the formation of α-acetolactate from pyruvate by the acetolactate synthase[34,35] (Fig. 5d). In animals, this pathway has been lost and branched-chain amino acids are essential nutrients[46]. Nevertheless, a homologous gene, the

**Fig. 4 | Dimethylglyoxal is formed from glucose and is increased under hypoxic conditions. a** [13]C-labeling in 3-deoxyglucosone (3-DG), glyoxal (Gx), methylglyoxal (MG), and dimethylglyoxal (DMG) in medium after treatment of mouse brain endothelial (bEnd.3) cells with [13C_6]-glucose for 24 h ($n = 3$). **b, c** Left and middle panels, time course of partially or fully [13]C-labeled MG and DMG in medium after adding [13C_6]-glucose to bEnd.3 cells and changing to hypoxic conditions as indicated (descriptive only). Right panels, fully [13]C-labeled MG and DMG under normoxic or hypoxic conditions at time point 360 min in an independent experiment ($n = 8$). For [13C_2]MG and [13C_2]DMG the position of [13]C in the molecule is unclear. Dicarbonyl levels were normalized to the parallel 360-min normoxia group. **d** Timeline of glucose injection, occlusion of the distal middle cerebral artery (MCAO), and sample collection. Part of the image was created with BioRender.com,

released under a Creative Commons Attribution-NonCommercial-NoDerivs 4.0 International license. **e** DMG levels in mouse plasma after MCAO and administration of vehicle ($n = 7$) or glucose ($n = 8$). **f** DMG levels in ipsi- and contralateral brain hemispheres after MCAO and administration of vehicle ($n = 7$ (30 min), $n = 5$ (ipsi, 50 min), $n = 6$ (contra, 50 min)), or glucose ($n = 8$ (30 min), $n = 8$ (ipsi, 50 min), $n = 7$ (contra, 50 min)). Values are means ± SEM. Statistical comparison by two-tailed unpaired $t$ test (**a, c**), two-tailed Mann–Whitney $U$ test (**b**) or by 2-way repeated-measures ANOVA (**e**) and mixed-effects analysis (REML, **f**) followed by Sidak's post hoc tests. *$p < 0.05$; **$p < 0.01$; ***$p < 0.001$. Detailed information on the test statistics is provided in Supplementary Table 5. Source data are provided as a Source Data file.

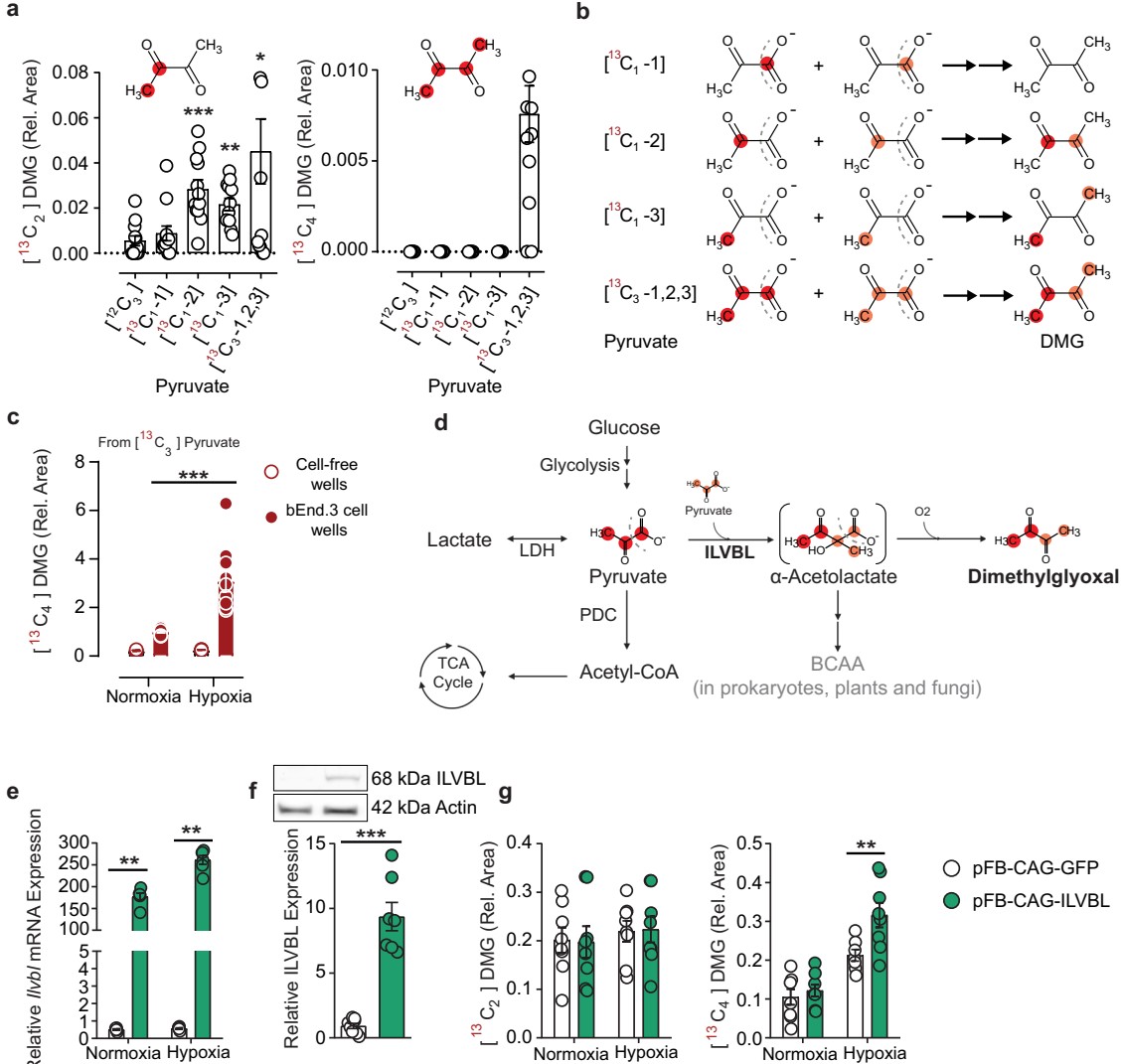

**Fig. 5 | Dimethylglyoxal production from pyruvate in endothelial cells is facilitated by ILVBL. a** Enrichment of [13C_2]-dimethylglyoxal ([13C_2]DMG) and [13C_4] DMG in the medium after treatment of mouse brain endothelial (bEnd.3) cells with [13C_1-1]-, [13C_1-2]-, [13C_1-3]-, or [13C_3-1,2,3]-pyruvate compared to native [12C_2]-pyruvate ($n = 12$). For [13C_2]DMG the position of [13]C in the molecule is unclear. **b** Proposed reaction scheme of [13C_2]DMG and [13C_4]DMG formation from two molecules of pyruvate with defined [13]C-labeling; created with BioRender.com, released under a Creative Commons Attribution-NonCommercial-NoDerivs 4.0 International license. **c** [13C_4]DMG in medium 6 hours after treatment with [13C_3]-pyruvate with ($n = 10$) or without ($n = 4$) bEnd.3 cells. **d** Proposed metabolic pathway to DMG. The unstable α-acetolactate in parenthesis is known as an intermediate product of the ilvB bacterial acetolactate synthase-like (ILVBL) ortholog in plants and bacteria. LDH lactate dehydrogenase, TCA tricarboxylic acid, PDC pyruvate dehydrogenase

complex, BCAA branched-chain amino acids. **e, f** *Ilvbl* mRNA ($n = 6$) and ILVBL protein levels ($n = 7$–8) 24 h after transfecting bEnd.3 cells with the ILVBL-expressing plasmid pFB-CAG-ILVBL or the control plasmid pFB-CAG-GFP. Protein levels were detected by Western blotting; a representative blot is shown in **f. g** [13C_2] DMG and [13C_4]DMG in the medium of control or ILVBL-overexpressing bEnd.3 cells, incubated for 6 h under normoxic or hypoxic conditions with [13C_6]-glucose ($n = 7$–8). Values are means ± SEM. Statistical comparison by Kruskal–Wallis followed by Dunn's post hoc tests (**a**), Scheirer-Ray-Hare test followed by two-tailed Mann–Whitney $U$ post hoc tests and Bonferroni-Holm correction (**c, e**), two-tailed Mann–Whitney $U$ test (**f**) or 2-way ANOVA followed by Tukey's post hoc tests (**g**). *$p < 0.05$; **$p < 0.01$; ***$p < 0.001$. Detailed information on the test statistics is provided in Supplementary Table 5. Source data are provided as a Source Data file.

bacterial acetolactate synthase-like (*Ilvbl*) is expressed in mammals[47] suggesting that it might be involved in the production of dimethylglyoxal. By transfecting bEnd.3 cells with an expression plasmid, we increased *Ilvbl* expression at the mRNA and protein level (Fig. 5e, f). In parallel to the higher *Ilvbl* expression, we observed higher $[^{13}C_4]$-dimethylglyoxal levels under hypoxic conditions (Fig. 5g), while $[^{13}C_2]$-methylglyoxal and $[^{13}C_3]$-methylglyoxal remained unchanged (Supplementary Fig. 3a). These data suggest that ILVBL contributes to dimethylglyoxal production in brain endothelial cells. The dependence of dimethylglyoxal on ILVBL explains the difference between methylglyoxal and dimethylglyoxal production, compatible with the concept of distinct pathways leading to diabetes complications, depending on whether glucose or lactate/pyruvate is the major energy source.

To explore the role of ILVBL for dimethylglyoxal production in vivo, we investigated $Ilvbl^{-/-}$ mice (Supplementary Fig. 3b). The animals are fertile and do not show an obvious phenotype. Under basal conditions, dimethylglyoxal concentrations in plasma, brain, and peripheral organs did not differ between $Ilvbl^{+/+}$ and $Ilvbl^{-/-}$ mice (Supplementary Fig. 3c–g). To stimulate dimethylglyoxal production, we induced acute hyperglycemia by administering $[^{13}C_6]$-glucose and subjected the mice to sham or MCAO treatment (Fig. 6a). In accordance with previous results (Fig. 4f), levels of unlabeled, $[^{13}C_2]$- or $[^{13}C_4]$-dimethylglyoxal were higher in the ischemic ipsilateral hemisphere and after MCAO than after sham treatment (Fig. 6b and Supplementary Fig. 3h). Notably, the increase in the ischemic ipsilateral hemisphere versus sham control was more pronounced with dimethylglyoxal than with methylglyoxal (unlabeled dimethylglyoxal, 3.4-fold, Fig. 6b; unlabeled methylglyoxal, 1.8-fold, Supplementary Fig. 3i). Importantly, *Ilvbl* deficiency mitigated the rise of unlabeled and $[^{13}C_2]$-dimethylglyoxal in the ischemic ipsilateral side of the brain (Fig. 6b). In contrast, the stroke-induced rise of labeled dimethylglyoxal in plasma (Fig. 6c) or of methylglyoxal and glyoxal in plasma or brain was not significantly affected by the *Ilvbl* deficiency (Supplementary Fig. 3i, j). Despite the lower dimethylglyoxal levels in the ischemic brain, $Ilvbl^{-/-}$ mice had similar infarct volumes and blood-brain barrier properties as $Ilvbl^{+/+}$ mice 2 days after MCAO (Supplementary Fig. 3k).

Upon STZ administration (Fig. 6d), $Ilvbl^{-/-}$ mice developed similar hyperglycemia as $Ilvbl^{+/+}$ littermates (Supplementary Fig. 3l). In parallel to glucose, dimethylglyoxal plasma levels increased in diabetic mice, as did those of glyoxal and 3-deoxyglucosone but methylglyoxal was not affected by diabetes (Fig. 6e and Supplementary Fig. 3m). While in plasma the dimethylglyoxal levels did not differ between $Ilvbl^{+/+}$and $Ilvbl^{-/-}$ animals, the brain levels were significantly reduced in $Ilvbl^{-/-}$ mice, whereas no changes were observed for glyoxal and methylglyoxal (Fig. 6f and Supplementary Fig. 3n). Thus, ILVBL contributes to dimethylglyoxal formation in the brain under hyperglycemic conditions, but the exact mechanism remains to be defined.

### Effects of dimethylglyoxal mimic diabetic complications
Dimethylglyoxal is a product of cellular pyruvate metabolism. To investigate its effects, we administered dimethylglyoxal to mice (Fig. 7a), using a single dose of 100 mg/kg (i.p.) that is well below the $LD_{50}$ of dimethylglyoxal[48]. After administering dimethylglyoxal, plasma concentrations were still elevated in the range of diabetic mice after 24 h (Fig. 7b, Supplementary Fig. 4a, cf. Fig. 1e, g). After 3 h, brain concentrations were raised, indicating that dimethylglyoxal is able to cross the blood–brain barrier (Fig. 7b, Supplementary Fig. 4a). The fast disappearance of the free compound from blood and brain could be due to its reaction with macromolecules. α-Dicarbonyls react mainly with side chains of basic amino acids[10,49]. Therefore, we searched for potential reaction products of dimethylglyoxal after its injection. Based on precursor mass, fragments in $ms^2$ spectra and retention time, compared to the labeled standard d4-Nε-carboxyethyllysine (d4-CEL), we were able to identify a compound that can be formed as reaction

product of dimethylglyoxal and lysine, $N_\varepsilon$-3-hydroxy-2-butanonelysine (HBL) (Fig. 7c). Even without dimethylglyoxal administration, the dimethylglyoxal adduct HBL was detected in brains of acutely hyperglycemic mice. Interestingly, HBL levels were significantly lower in $Ilvbl^{-/-}$ mice than in $Ilvbl^{+/+}$ littermate controls (Fig. 7d), confirming the involvement of ILVBL in dimethylglyoxal formation in the brain and indicating that dimethylglyoxal is able to react with proteins or amino acids in vivo.

Since methylglyoxal induces oxidative stress[7], we wanted to know whether dimethylglyoxal has similar properties in cells that mediate neurological complications of hyperglycemia. We determined the production of reactive oxygen species (ROS) with a fluorescent dye in brain endothelial bEnd.3 cells, primary astrocytes, hippocampal neuronal HT22 cells, and M1- or M2-polarized bone-marrow-derived macrophages (BMDM). The latter contribute to the effects of α-dicarbonyls in hyperglycemic stroke[14]. Indeed, a 10-min treatment with dimethylglyoxal strongly induced ROS in all cell types (Fig. 8a–c and Supplementary Fig. 4b–d). Already low concentrations (≥20 μM) of dimethylglyoxal stimulated ROS formation in bEnd.3 cells (Supplementary Fig. 4b). Dimethylglyoxal was even more effective than methylglyoxal in bEnd.3 cells, astrocytes, and M1- or M2-polarized BMDM (Fig. 8a, b and Supplementary Fig. 4c, d). Thus, in addition to modifying lysine residues to HBL, dimethylglyoxal is a strong inducer of oxidative stress.

As inflammatory changes are a hallmark of diabetes, we wondered whether elevated blood and brain concentrations of dimethylglyoxal would also induce inflammation. To test for neuroinflammation, we counted Iba1+ microglia and brain macrophages after dimethylglyoxal injection (Fig. 7a) and found increased numbers in the visual, motor, and somatosensory cortex (Fig. 8d, e and Supplementary Fig. 4e). In vitro, dimethylglyoxal treatment of bEnd.3 cells for 24 h induced the expression of the chemokine *Ccl2* and reduced the anti-inflammatory cytokine *Tgfβ2* at the mRNA level, suggesting that endothelial cells create a pro-inflammatory environment upon dimethylglyoxal exposure (Fig. 8f). In neuronal HT22 cells dimethylglyoxal stimulated the expression of the pro-inflammatory and pro-apoptotic genes *Pla2g4a*, *Ptges*, *Ptgs2*, and *Bim* that are involved in ischemic brain damage[50,51] (Fig. 8g). Furthermore, dimethylglyoxal treatment for 48 h promoted pro-inflammatory differentiation of BMDM, indicated by increased *Ccl2*, *Tnf*, *Il-1b*, and *Il-1a* expression (Supplementary Fig. 4f). In contrast, dimethylglyoxal reduced polarization of BMDM toward an anti-inflammatory phenotype, reflected by decreased *Retnla* and *Arg1* expression (Supplementary Fig. 4g).

Dysfunction of the blood–brain barrier is well known in diabetes[52,53] and a key factor in the development of vascular dementia[54,55]. Occludin, a tight junction protein in the blood-brain barrier, is downregulated in diabetes[56,57]. Interestingly, occludin staining was decreased in endothelial cells of the hippocampus and cortex 3 h after dimethylglyoxal injection (Fig. 8h, i and Supplementary Fig. 4h, i). After 24 h, occludin levels had recovered. In primary brain endothelial cells (PBEC), dimethylglyoxal lowered occludin levels after 3 h, as in vivo, whereas methylglyoxal had no effect on occludin levels (Supplementary Fig. 4j), arguing against the possibility that methylglyoxal could mediate the dimethylglyoxal effect.

To investigate how a prolonged increase in dimethylglyoxal concentrations as previously seen in diabetic mice affects the cerebrovascular system, we aimed to mimic the endogenous dimethylglyoxal production by oral administration of the compound. In pilot experiments, we were able to detect $[^{13}C_4]$-dimethylglyoxal in the plasma and brain after oral administration of $[^{13}C_4]$-dimethylglyoxal to 3 mice, indicating that dimethylglyoxal is absorbed in the gastrointestinal tract and even reaches the brain (Fig. 9a, Supplementary Fig. 5a). On a side note, we could not detect labeled methylglyoxal or glyoxal in these experiments suggesting that dimethylglyoxal does not act as a precursor of other α-dicarbonyls. Encouraged by the intestinal

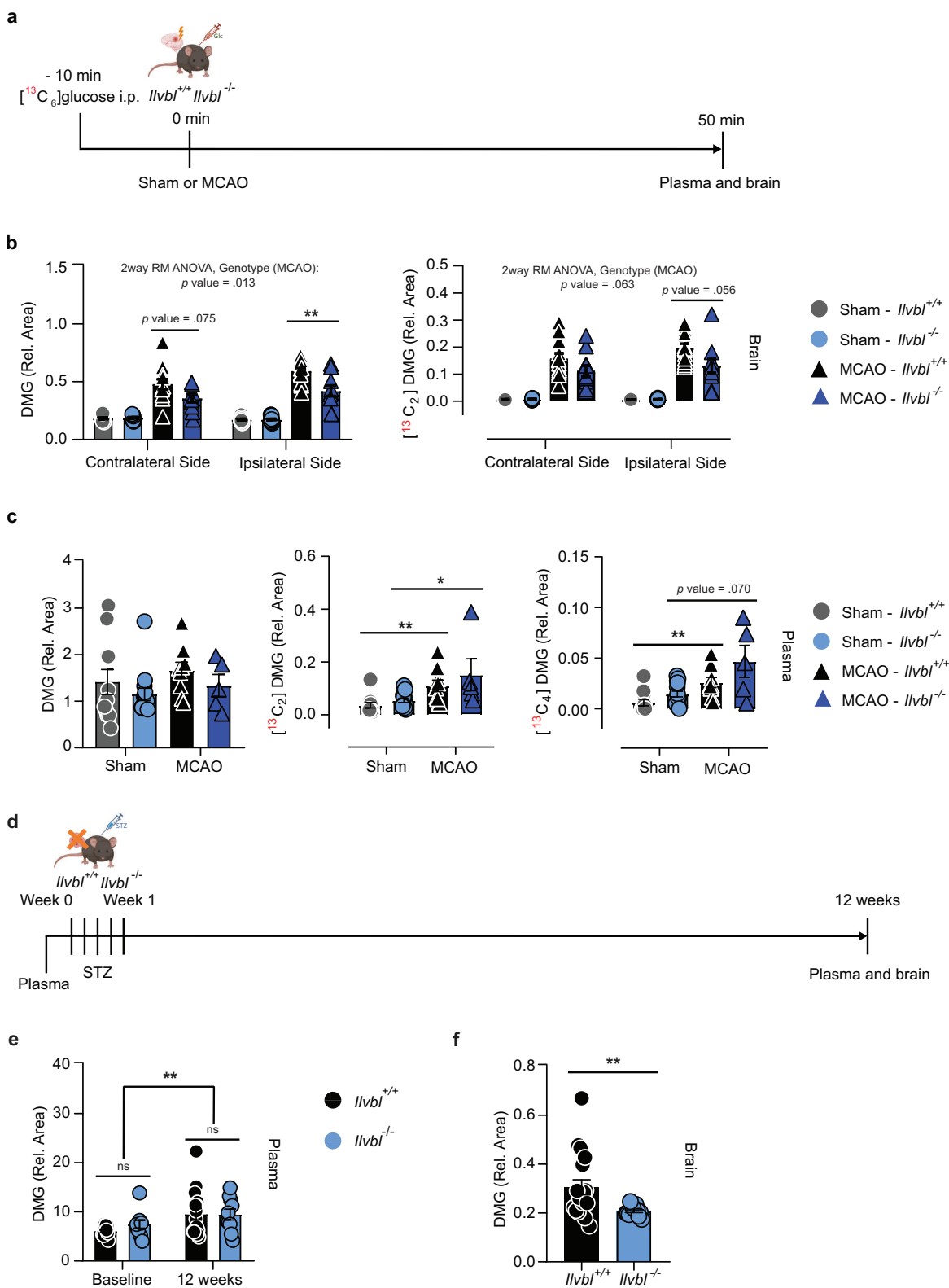

**Fig. 6 | ILVBL is involved in the generation of dimethylglyoxal in the brain of mice with hyperglycemic stroke or diabetes. a** Timeline of [$^{13}C_6$]-glucose injection, sham surgery or occlusion of the distal middle cerebral artery (MCAO), and sample collection. Part of the image was created with BioRender.com, released under a Creative Commons Attribution-NonCommercial-NoDerivs 4.0 International license. **b** Dimethylglyoxal (DMG) and [$^{13}C_2$]DMG in the ipsi- and contralateral hemispheres of *Ilvbl*$^{+/+}$ or *Ilvbl*$^{-/-}$ mice, after sham surgery ($n = 14$ (*Ilvbl*$^{+/+}$), 17 (*Ilvbl*$^{-/-}$)) or MCAO ($n = 12$ (*Ilvbl*$^{+/+}$), $n = 10$ (*Ilvbl*$^{-/-}$)). **c** DMG, [$^{13}C_2$]DMG, and [$^{13}C_4$]DMG in plasma of *Ilvbl*$^{+/+}$ or *Ilvbl*$^{-/-}$ mice after sham surgery ($n = 11$ (*Ilvbl*$^{+/+}$), $n = 14$ (*Ilvbl*$^{-/-}$)) or MCAO ($n = 9$ (*Ilvbl*$^{+/+}$), $n = 5$ (*Ilvbl*$^{-/-}$)). **d** Timeline of plasma collection before the treatment, streptozotocin (STZ)-induced diabetes, and sample collection. **e** DMG in plasma of *Ilvbl*$^{+/+}$ ($n = 16$) or *Ilvbl*$^{-/-}$ ($n = 10$) mice at baseline and 12 weeks after STZ treatment. **f** DMG in brains of *Ilvbl*$^{+/+}$ ($n = 18$) or *Ilvbl*$^{-/-}$ ($n = 12$) mice 12 weeks after STZ treatment. Values are means ± SEM. Statistical comparison by 2-way repeated-measures ANOVA followed by Sidak's post hoc tests (**b**, **e**), Scheirer-Ray-Hare test followed by two-tailed Mann–Whitney $U$ post hoc tests and Bonferroni-Holm correction (**c**) or two-tailed Mann–Whitney $U$ test (**f**). *$p < 0.05$; **$p < 0.01$. Detailed information on the test statistics is provided in Supplementary Table 5. Source data are provided as a Source Data file.

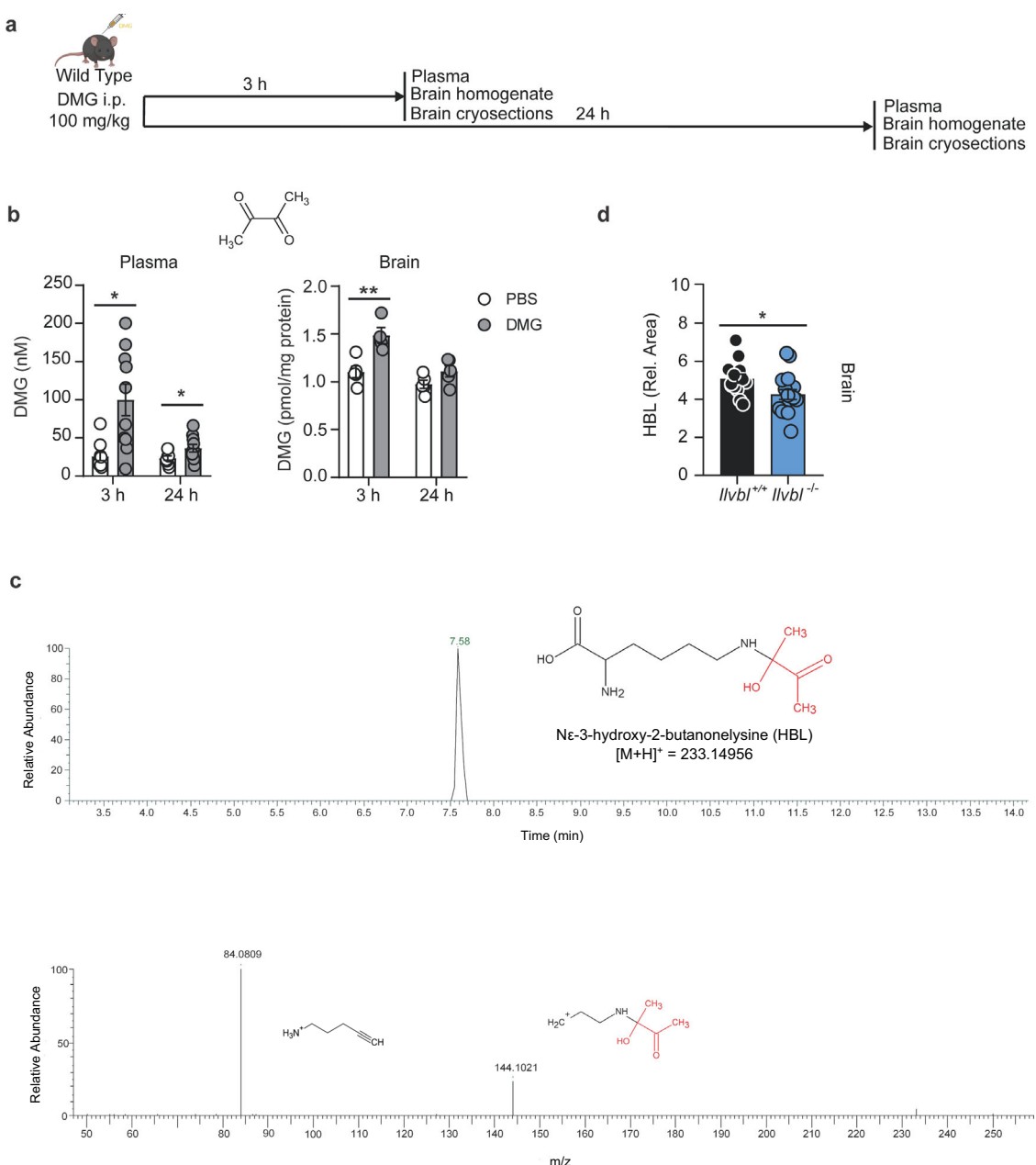

**Fig. 7 | Dimethylglyoxal reacts with lysine to the glycated amino acid $N_\varepsilon$-3-hydroxy-2-butanonelysine (HBL) in vivo. a** Timeline of dimethylglyoxal (DMG) administration and sample collection. Part of the image was created with BioRender.com, released under a Creative Commons Attribution-NonCommercial-NoDerivs 4.0 International license. **b** DMG concentration in plasma ($n = 9$–10) and brain ($n = 4$–5) 3 h or 24 h after i.p. DMG injection. **c** Representative chromatogram and fragmentation mass spectrum of HBL (retention time = 7.58 min; $m/z$ = 233.150) in the brain of a DMG-treated mouse. **d** Levels of HBL in brains of $Ilvbl^{+/+}$ ($n = 14$) or $Ilvbl^{-/-}$ ($n = 17$) mice with acute hyperglycemia. Values are means ± SEM. Statistical comparison by Scheirer-Ray-Hare test followed by two-tailed Mann–Whitney $U$ post hoc tests and Bonferroni–Holm correction (**b**, plasma), 2-way ANOVA followed by Tukey's post-hoc tests (**b**, brain) or two-tailed unpaired $t$ test (**d**). *$p < 0.05$; **$p < 0.01$. Detailed information on the test statistics is provided in Supplementary Table 5. Source data are provided as a Source Data file.

absorption, we administered dimethylglyoxal (100 mg kg$^{-1}$day$^{-1}$) for 14 weeks via the drinking water (Fig. 9b). With this regimen, after 6 and 14 weeks dimethylglyoxal plasma concentrations were increased to a similar level as in STZ-treated mice (Fig. 9c, d), while methylglyoxal was not changed by the chronic dimethylglyoxal treatment (Supplementary Fig. 5b). After 14 weeks, the dimethylglyoxal adduct HBL was also significantly increased in plasma (Fig. 9d). Although orally administered [$^{13}C_4$]-dimethylglyoxal reached the brain (Fig. 9a), the chronic administration did not change brain concentrations of dimethylglyoxal, in contrast to STZ treatment (Supplementary Fig. 5c, Fig. 1f, and Supplementary Fig. 1d).

Mice treated with STZ or dimethylglyoxal showed mostly normal mobility and explorative and anxiety-like behavior in the open field and the elevated plus maze (Supplementary Fig. 5d–g). In STZ-treated mice, locomotion in the open field was transiently reduced (Supplementary Fig. 5e). Although the treatment groups did not differ in the open field, it was interesting to note that dimethylglyoxal brain concentrations were inversely correlated with the speed of movement in the central zone and positively correlated with the time the mice were immobile in the open field (Supplementary Fig. 5h, i). In both groups spatial working memory as tested by the Y-maze was intact (Supplementary Fig. 5j). In the Barnes maze, both STZ and dimethylglyoxal

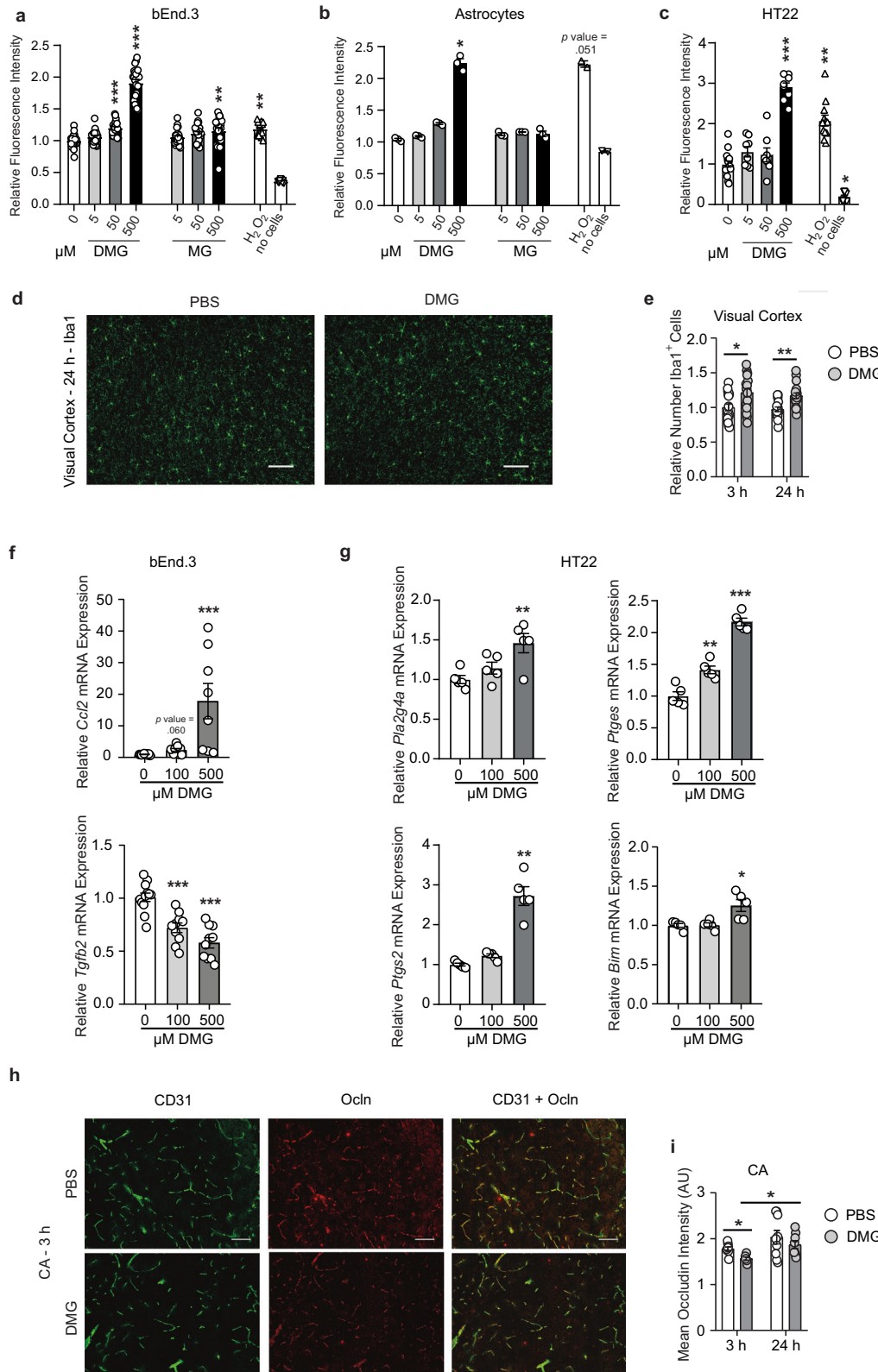

treatment groups did not differ from controls in the number of primary errors but more STZ-treated than control mice lacked a strategy to search the target hole in the probe trial on day 5 (Supplementary Fig. 5k, l). Interestingly, dimethylglyoxal-treated mice showed signs of mild cognitive decline in the object place recognition test (OPRT). After a 1-h delay, they identified the replaced object less efficiently than vehicle-treated controls (Fig. 9e). In this, they mimicked the phenotype

of diabetic mice, twelve weeks after diabetes was induced with STZ (Figs. 1a and 9f). Overall, STZ and dimethylglyoxal treatment induced mild behavioral changes in line with subtle cognitive deficits in diabetes patients[1].

Cognitive changes induced by chronic dimethylglyoxal administration or by diabetes were accompanied by increased oxidative stress in the cerebrovascular system, as indicated by enhanced 4-HNE

**Fig. 8 | Dimethylglyoxal induces oxidative stress, inflammation and reduces occludin levels. a–c** Induction of reactive oxygen species (ROS) in mouse brain endothelial cells (bEnd.3; **a**; $n = 22\text{-}24$), primary astrocytes (**b**; $n = 3$), neuronal HT22 cells (**c**; $n = 8$ except $n = 16$ for 0 μM DMG) by treatment with dimethylglyoxal (DMG), methylglyoxal (MG) or hydrogen peroxide ($H_2O_2$; 10 μM; $n = 16$ (**a**), $n = 2$ (**b**), $n = 12$ (**c**)) for 10 min. ROS production was detected with the dye chloromethyl 2′,7′-dichlorodihydrofluorescein diacetate (CM-$H_2$DCFDA). **d, e** Representative microscope images of Iba1+ microglia and brain macrophages in visual cortex after intraperitoneal injection of 100 mg/kg DMG (Fig. 7a) and quantification after 3 ($n = 17\text{-}18$) and 24 h ($n = 16\text{-}18$). **f, g** mRNA expression of pro- and anti-inflammatory or pro-apoptotic genes in bEnd.3 (**f**; $n = 8\text{-}12$) or HT22 cells (**g**; $n = 5$) treated with

DMG for 24 hours. **h** Representative microscopy images of occludin (Ocln) and cluster of differentiation 31 (CD31) immunohistochemistry in *cornu Ammonis* (CA) 3 h after DMG injection. **i** Quantification of Ocln intensity in brain vessels restricted to CD31+ signal in CA 3 h ($n = 6\text{-}9$) and 24 h ($n = 8\text{-}9$) after DMG injection. AU, arbitrary units. Values are means ± SEM. Statistical comparison by Kruskal-Wallis followed by Dunn's post hoc tests (**a–c**, **f** (*Ccl2*), **g** (*Ptgs2*, *Bim*)), Scheirer-Ray-Hare test followed by two-tailed Mann−Whitney *U* post hoc tests and Bonferroni-Holm correction (**e**, **i**) or one-way ANOVA followed by Dunnett's post hoc tests (**f** (*Tgfb2*), **g** (*Pla2g4a*, *Ptges*)), *$p < 0.05$; **$p < 0.01$. ***$p < 0.001$. Scale bar 100 μm. Detailed information on the test statistics is provided in Supplementary Table 5. Source data are provided as a Source Data file.

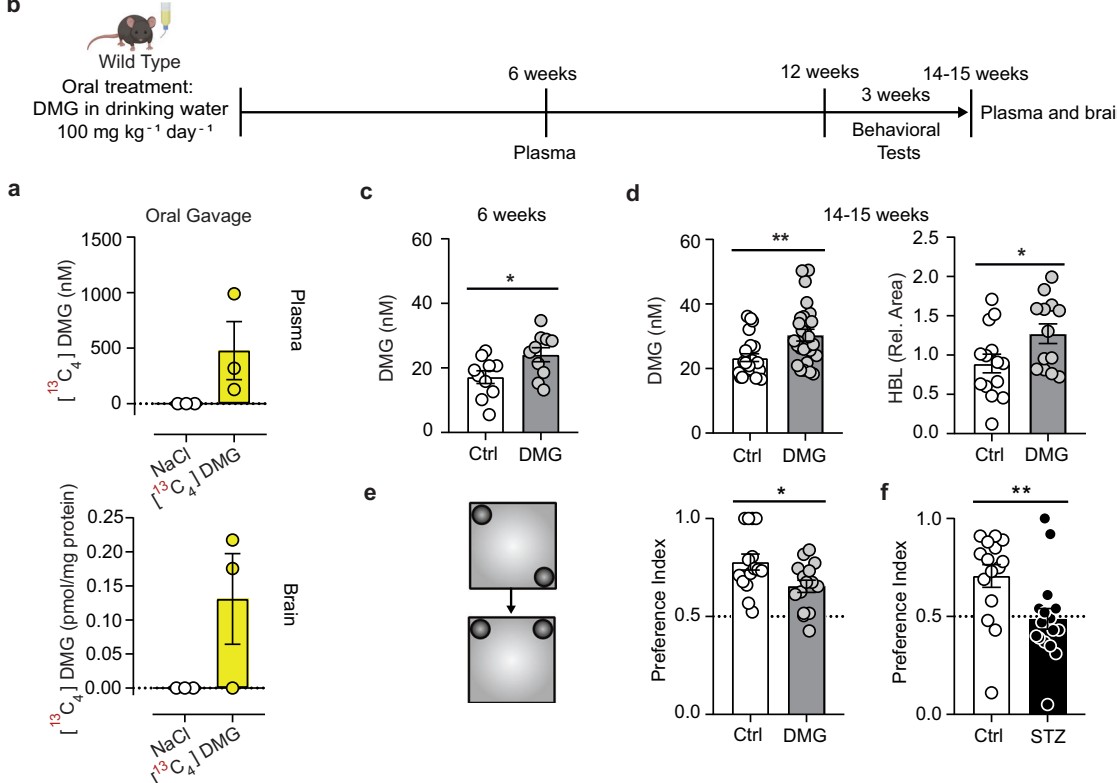

**Fig. 9 | Long-term treatment with dimethylglyoxal induces cognitive decline. a** Concentrations of [${}^{13}C_4$]-dimethylglyoxal (DMG) in plasma and brain 20 min after gavage of mice with NaCl or [${}^{13}C_4$]DMG (5 mg/kg; $n = 3$). **b** Timeline of treatment with DMG in drinking water, plasma collection after 6 weeks, behavioral tests, and sample collection after 14−15 weeks. The mouse image was created with BioRender.com, released under a Creative Commons Attribution-NonCommercial-NoDerivs 4.0 International license. **c** DMG concentrations in plasma after 6 weeks of oral DMG treatment ($n = 10$). **d** DMG ($n = 24\text{-}26$) and $N_\varepsilon$-3-hydroxy-2-

butanonelysine (HBL; $n = 13\text{-}14$) plasma levels after 14−15 weeks of oral DMG treatment. **e, f** Performance of mice in an object place recognition test (OPRT) after 12 weeks of oral DMG treatment (**e**; $n = 15$) or 12 weeks after streptozotocin (STZ) treatment (**f**; $n = 15\text{-}17$; Fig. 1a). Values are means ± SEM. Statistical comparison by two-tailed unpaired *t* test (**c**, **d** (HBL), **e**, **f**) or two-tailed Mann−Whitney *U* test (**d** (DMG)). *$p < 0.05$; **$p < 0.01$. Detailed information on the test statistics is provided in Supplementary Table 5. Source data are provided as a Source Data file.

staining of vessels (Fig. 10a–d). At the same time, dimethylglyoxal reduced occludin levels in endothelial cells and increased IgG extravasation into the brain parenchyma, demonstrating that dimethylglyoxal impaired the blood-brain barrier (Fig. 10e-g). Considering the pro-inflammatory properties of dimethylglyoxal, we measured plasma levels of cytokines. In accordance with its effects in vitro, dimethylglyoxal administration elevated Il-1α concentrations in plasma (Fig. 10h). In the hippocampus, astrocytes express the receptor for Il-1α (Il-1R1) and respond to IL-1[58]. In line with this, dimethylglyoxal treatment increased GFAP staining in the hippocampus, reflecting its pro-inflammatory properties (Fig. 10i and Supplementary Fig. 5m).

## Discussion

We have identified the α-dicarbonyl dimethylglyoxal as a product of cellular pyruvate metabolism. This compound, also called diacetyl, has long been known as a microbial metabolite and flavor[59,60]. In the food industry, occupational exposure to high concentrations caused bronchiolitis obliterans, indicating that dimethylglyoxal is biologically active[61]. Even without exposure to high doses, dimethylglyoxal has been detected in mammalian tissues and samples[27,62] but its origin and function remained unclear.

Our data demonstrates that whilst dimethylglyoxal is remotely derived from glucose, there are several important differences to other α-dicarbonyls. 3-deoxyglucosone, glyoxal, and methylglyoxal

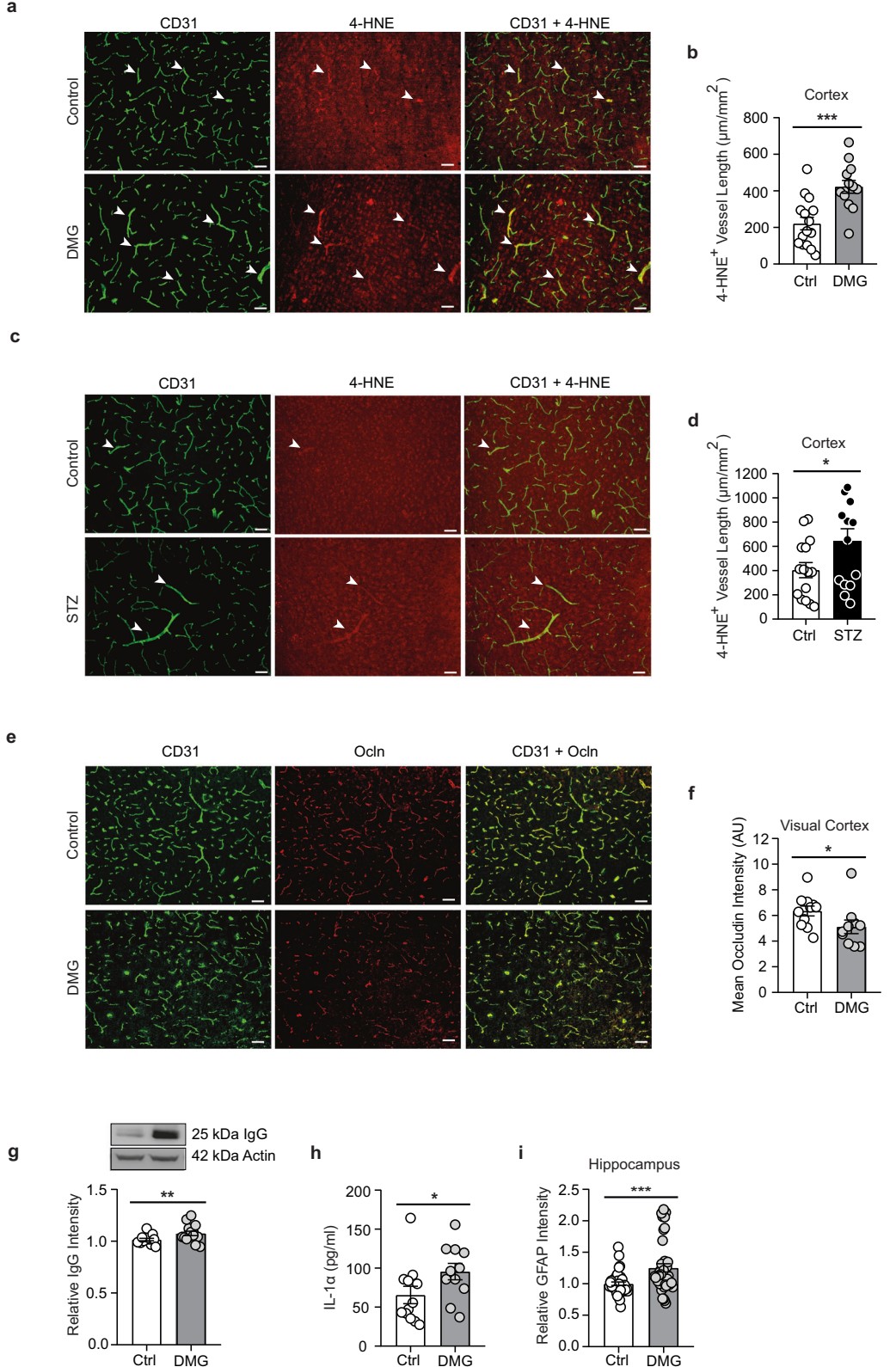

**Fig. 10 | Long-term treatment with dimethylglyoxal damages the blood-brain barrier. a**, **b** Representative images of 4-hydroxynonenal (4-HNE)- and CD31-stained cortical brain sections and length of 4-HNE⁺ CD31⁺ vessels, reflecting oxidative stress in cerebral endothelial cells of dimethylglyoxal (DMG)-treated mice (*n* = 13–15). **c**, **d** Representative images of 4-HNE- and CD31-stained cortical brain sections and length of 4-HNE⁺ CD31⁺ vessels in streptozotocin (STZ)-treated mice (*n* = 14–15). **e**, **f** Representative images of occludin (Ocln)- and CD31-immunostained cortical brain sections and quantification of Ocln intensity in brain vessels (*n* = 10–11). AU, arbitrary units. **g** Western blot analysis of immunoglobulin G (IgG) light chain extravasation into brain tissue (*n* = 14–15). A representative blot is shown. **h** Plasma concentrations of interleukin-1 alpha (IL-1α; *n* = 11–12). **i** Quantification of mean glial fibrillary acidic protein (GFAP) immunohistological staining intensity (*n* = 19–20). Values are means ± SEM. Statistical comparison by two-tailed unpaired t test (**b**, **d**) or two-tailed Mann–Whitney *U* test (**f**–**i**). **p* < 0.05; ***p* < 0.01; ****p* < 0.001. Scale bar 100 μm. Detailed information on the test statistics is provided in Supplementary Table 5. Source data are provided as a Source Data file.

are formed by autooxidation of glucose or as by-products of the first phase of glycolysis. In contrast, we have been able to demonstrate the pyruvate metabolism as a source of dimethylglyoxal formation; after decarboxylation, the C2 and C3 atoms of two pyruvate molecules condense to form dimethylglyoxal. Apparently, PDC does not mediate this reaction because hypoxia, which is known to inhibit PDC[44,45], increased dimethylglyoxal production and activation of PDC by dichloroacetate did not stimulate dimethylglyoxal production. We propose that upon PDC inhibition by hypoxia, pyruvate builds up and is redirected to other pathways. In plants and microorganisms, acetolactate synthase converts pyruvate into acetolactate, an intermediate for the synthesis of branched-chain amino acids[34]. Acetolactate spontaneously decarboxylates and oxidizes to dimethylglyoxal. Animals have lost the capacity to synthesize branched-chain amino acids[46], but the gene *Ilvbl* that is homologous to yeast acetolactate synthase is conserved[47]. The present study indicates that ILVBL has retained its evolutionarily old function in dimethylglyoxal formation. We found lower dimethylglyoxal concentrations in brains of *Ilvbl*[-/-] than *Ilvbl*[+/+] mice after STZ treatment or hyperglycemic stroke. In addition to ILVBL, other not yet identified pathways contribute to the formation of dimethylglyoxal as *Ilvbl*[-/-] mice had still significant dimethylglyoxal concentrations in brain and unchanged levels in blood. This may explain why *Ilvbl*[-/-] mice were not protected against hyperglycemic stroke concerning infarct size or blood-brain barrier integrity.

The generation of dimethylglyoxal downstream of pyruvate suggests that cells consuming lactate and pyruvate as energy sources, such as neurons and brain endothelial cells, produce more dimethylglyoxal than others. α-Dicarbonyls are released from cells and can penetrate cell membranes[63], but their high reactivity with amino acid residues creates a gradient of concentrations and functional impact around the site of production[64]. Therefore, the formation of dimethylglyoxal from pyruvate may explain the paradox that neurons and other cell types, although they have largely outsourced glycolysis, are not protected from diabetic complications[65]. For the effects of α-dicarbonyls, it seems to be important that they are produced in different subcellular compartments. While glycolysis producing methylglyoxal takes place in the cytosol, in yeast pyruvate metabolism by acetolactate synthase is localized in mitochondria[66]. A similar mechanism in mammalian cells may contribute to mitochondrial dysfunction in diabetes[67]. However, a recent study reported that mammalian ILVBL is localized in the endoplasmic reticulum (ER)[68], possibly linking dimethylglyoxal formation to ER stress in diabetes[69]. It must be noted that the high reactivity and the resulting limited distribution in biological samples pose a problem for the modeling of endogenous production by administration of exogenous dimethylglyoxal.

As a glucose metabolite, dimethylglyoxal was elevated in blood and brain tissue in experimental and clinical diabetes. In line with this, a previous study has detected higher dimethylglyoxal concentrations in the exhaled air of subjects with a pathological oral glucose tolerance test; the authors attributed dimethylglyoxal production to bacteria in the mouth[70]. Our data would argue that dimethylglyoxal was derived from human metabolism. Measuring its concentrations in the exhaled air may represent a simple and noninvasive way to monitor α-dicarbonyl stress in diabetes. The enhanced production in diabetes is remarkable because dimethylglyoxal is a highly reactive compound. Reactivity of α-dicarbonyls is due to the non-hydrated form[71]. Interestingly, in aqueous solutions the fraction of non-hydrated α-dicarbonyl compound increases with the number of methyl groups (glyoxal, 0.005%; methylglyoxal, 1%; dimethylglyoxal, 30%)[71,72], suggesting that concentrations of dimethylglyoxal in the reactive non-hydrated form may exceed those of other α-dicarbonyls. As evidence for its high reactivity, we found the glycated amino acid HBL, an adduct of

dimethylglyoxal to lysine in the brain. In addition, dimethylglyoxal induced oxidative stress and upregulated inflammatory markers. Probably because of these effects, dimethylglyoxal impaired blood-brain barrier function and caused cognitive defects in mice, resembling the cognitive deficits observed in mice with STZ-induced diabetes. Based on these data, we propose that dimethylglyoxal is an important factor in causing cognitive dysfunction in diabetes. Whether dimethylglyoxal is involved in other diabetes complications is still unclear.

The pathway leading to dimethylglyoxal synthesis has ramifications for potential approaches to reduce carbonyl stress. Attempts to activate glyoxalase, the enzyme that degrades methylglyoxal, may not succeed in combating the toxic effects of dimethylglyoxal in diabetes because dimethylglyoxal inhibits glyoxalase I and is degraded by other enzymes[73,74]. An attractive alternative option is metformin, which, by inhibiting the mitochondrial glycerol-3-phosphate dehydrogenase, shifts the ratio of lactate to pyruvate in the lactate direction[75]. The pathway of dimethylglyoxal synthesis from pyruvate that we have deciphered predicts that lowering pyruvate availability would reduce dimethylglyoxal production by ILVBL. In line with this, metformin plus insulin was more effective in lowering dimethylglyoxal in diabetes patients than was insulin. This mechanism may contribute to the superior efficacy of metformin in preventing diabetes complications[76]. Beyond the availability of pyruvate, ILVBL is a druggable enzyme, at least in other species[77], suggesting new treatment strategies against diabetic complications.

## Methods

### Study approval
The collection of clinical data and blood of humans took place upon written informed consent by the patients according to the Declaration of Helsinki following the approval of the local ethical committee of the University of Lübeck. The local animal welfare authorities (Ministerium für Energiewende, Landwirtschaft, Umwelt und ländliche Räume, Kiel, Germany) approved all mouse experiments.

### Diabetes patients
Serum samples of 100 patients with diabetes and 100 age- and sex-matched inpatients without any signs of diabetes mellitus were collected in the Medical Clinic I of the University Medical Center Schleswig-Holstein in Lübeck, Germany. Here, blood samples were incubated for 30 min at room temperature followed by centrifugation ($3000 \times g$, 10 min, 4 °C). Then, serum was stored at −20 °C until detection using LC-MS. Glucose concentrations were measured in serum samples using the Glucose-Glo™ Assay (Promega, USA) according to the manufacturer's instruction. The covariate sex had no significant effect on the dicarbonyl values and was not taken into account in the analysis.

### Mice
*Ilvbl*[-/-] mice (Ilvbl[em1(IMPC)Bay] null allele (C57BL/6 N), MGI:1351911) were kindly provided by the Knockout Mouse Phenotyping Project (KOMP2). We compared *Ilvbl*[-/-] mice with *Ilvbl*[+/+] littermate controls obtained by heterozygous breeding. For STZ experiments, male mice at an age of 6-10 weeks were used. MCAO experiments were performed in animals of both sexes. C57BL/6 N wild-type mice were ordered from Charles River Europe and male *db/db* (BKS(D)-*Lepr*[db]/JOrlRj, MGI:6293869) or heterozygous *db/+* littermates from Janvier Labs. Animals were kept in individually ventilated cages in groups of three to five under a 12-h light/dark cycle at 23 °C with *ad libitum* access to food (Altromin, #1314 unless stated otherwise) and water. According to the ARRIVE guidelines, investigators were blinded to treatment and genotype. Mice were randomly allocated to treatment groups. Mice were euthanized by transcardial perfusion under deep anesthesia or by decapitation under isoflurane anesthesia.

## Dimethylglyoxal treatment of mice

For acute dimethylglyoxal treatment, mice received injections of dimethylglyoxal (100 mg/kg in PBS, i.p.) or PBS and were sacrificed after 3 or 24 h. [$^{13}C_4$]-dimethylglyoxal (Eptes, Vevey, Switzerland) was administered by gavage. For chronic treatment, the drinking water of mice was supplemented with 500 mg/l dimethylglyoxal, resulting in a dosage of approximately 100 mg/kg/day with a daily water intake of 5 ml and 25 g body weight. After 6 weeks, dimethylglyoxal concentrations were measured in EDTA plasma collected from the tail vein. Behavioral tests were performed after 12 weeks and mice were sacrificed at week 14–15. Cytokines were detected in 25 µl plasma using the Milliplex® assay according to the manufacturer's instructions (Mouse high-sensitivity cell magnetic bead panel, Merck, MHSTCMAG-70k).

## Streptozotocin injection and dietary interventions

Mice received an i.p. injection of 50 mg/kg STZ in 50 mM sodium citrate buffer (pH 4.5) on five consecutive days. During this period, mice were provided with normal chow and 10% (w/v) sucrose to prevent hypoglycemia. Induction of chronic hyperglycemia was verified after 2 and 14–15 weeks using glucose test stripes (Ascensia Elite XL, Roche Diagnostics). Finally, mice were deeply anesthetized and half of the brain (sagittal) as well as EDTA plasma were collected and processed for LC-MS/MS analysis (see below). For immunostainings, the other half of the brain was frozen on dry ice. Insulin in plasma was detected using a Rat/Mouse Insulin ELISA kit (MERCK, EZRMI-13K) according to the manufacturer's instructions. Mice were excluded if blood glucose did not rise to >11.1 mM 6 weeks after STZ administration.

For ketogenic diet, mice were adapted to the new food using first a 25:75 and then a 50:50 mixture of ketogenic diet (80% fat, 8% protein, 1.3% carbohydrates, Sniff E15149-307) or control diet (5% fat, Sniff E15051-047) with normal chow for 5 days each until food intake was normalized. Then, mice were treated for 6 weeks with pure ketogenic or control diet. Glucose and pH in urine were determined on day 10 and 56 using test stripes (Multistix®10 SG, Siemens).

As a model of diet-induced obesity, C57BL/6J mice were fed with high-fat diet (Research diets, D12492) or control diet (Teklad Global, 2018, 18% protein rodent diet, irradiated) for 12 weeks and euthanized by decapitation.

## LC-MS/MS analysis

Deuterated methylglyoxal (d4-MG) was synthesized, as previously described[78], using d6-acetone. The concentration and purity of the d4-MG stock solutions were determined from 1H, 2H and 13 C NMR spectra acquired at 298 K using a Bruker Avance II NMR spectrometer. The purities of the d4-MG stock solutions were 60-65% with the major contaminants being d3-acetate and d6-acetone.

The internal standards d4-MG (400 nM, 0.2 µl per µl plasma) and $^{13}C_3$-pyruvate (200 µM, 0.2 µl per µl plasma) were added to plasma (25 µl or 60 µl). Then, proteins were precipitated by adding ice-cold trichloroacetic acid (25% w/v, 0.4 µl per µl plasma) followed by adding LC-MS grade water (0.8 µl per µl plasma). Brain extracts (25 µl or 120 µl) or cell culture supernatants (45 µl) were used without diluting water. Sample were mixed (4 °C, 1,000 rpm, 10 min) followed by centrifugation (20,000 × g, 10 min, 4 °C). The supernatant was transferred to a glass vial and the α-dicarbonyls were derivatized to their respective quinoxaline compounds for 4 h in the dark using 10–12 µl o-phenylenediamine or d8-o-phenylenediamine (CDN Isotopes, 0.5 – 1.0 mM) in HCl/diethylenetriaminepentaacetic acid (DETA-PAC, 0.5 mM).

For detecting glycated amino acids, 360 µl acetonitrile with 0.1% formic acid was added to 100 µl plasma and 180 µl acetonitrile with 0.1% formic acid to 50 µl brain homogenate. After adding the internal standard deuterated N$_\varepsilon$-(1-carboxyethyl)-L-lysine (d4-CEL, 400 nM in water, PolyPeptide Group, 10 µl for plasma and 5 µl for brain

homogenate), samples were mixed thoroughly and incubated for 30 min at −20 °C. After centrifugation (20,000 × g, 10 min, 4 °C), supernatant was transferred to glass vials for LC-MS detection.

Native α-dicarbonyls and glycated amino acids were quantitatively detected using the TSQ Endura triple quadrupole mass spectrometer (Thermo Scientific), while the Q-Exactive quadrupole-orbitrap hybrid mass spectrometer (Thermo Scientific) was used to detect $^{13}$C-labeled α-dicarbonyls and identify DMG-derived glycated amino acids. Both spectrometers were coupled to a Dionex Ultimate 3000 HPLC system (Thermo Scientific) and equipped with a heated-electrospray ionization (HESI-II) probe. Quinoxalines were separated on an Ascentis Express C18 column (100 mm×2.1 mm×5 µm, Supleco) or Cortecs®T3 (100 mm×2.1 mm×2.7 µm, Waters) for TSQ-Endura or Q-Exactive, respectively, while the XBridge BEH Amide column (100 mm×2.1 mm, 2.5 µm, Waters) was used for glycated amino acids in both cases. For chromatography, a gradient elution with 0.1% formic acid in water as mobile phase A and 0.1% formic acid in acetonitrile as mobile phase B with a flow rate of 0.2 ml/min or 0.4 ml/min for quinoxalines and glycated amino acids was used. For quinoxalines, an isocratic gradient of 90% A and 10% B for 10 min was applied. Afterwards, B increased to 100% between 10 and 11 min, which was followed by a washing step with 100% B between 11 and 15 min with a flow rate of 0.4 ml/min. Re-equilibration took place between 15 and 20 min back to 10% B. Following parameters were used in the positive ionization mode: ion spray voltage, 4600 V; vaporizer temperature, 100 °C; and ion transfer tube temperature, 300 °C. For glycated amino acids, chromatographic separation was realized using following gradient: 95% B from 0 to 3 min, followed by a decrease to 60% B from 3 to 6 min and to 30% B from 6 to 12 min. After washing at 30% B from 12 to 16 min, re-equilibration took place by increasing to 95% B from 16 to 25 min and keeping 95% for another 5 min. Following parameters were used in the positive ionization mode: ion spray voltage, 3500 or 4000 V; vaporizer temperature, 200 °C; and ion transfer tube temperature, 300 °C. Multiple reaction monitoring (MRM; TSQ-Endura) or full MS data-dependent MS$^2$ acquisition (top 10) in combination with parallel reaction monitoring (PRM, Q-Exactive) was applied to detect and identify the molecules. Absolute quantification was performed with an external calibration curve based on the ratio of the areas under the peaks to the internal standard. Relative quantification was based on the ratio to the internal standard only. For normalization of HBL, d4-CEL was used as internal standard. Specifications for detecting native α-dicarbonyls and glycated amino acids are listed in Supplementary Table 1, and for $^{13}$C-labeled molecules in Supplementary Table 2.

The method for detecting lactate using the Q-Exactive mass spectrometer (Thermo Scientific) has been described previously[79]. Briefly, 200 µl or 400 µl acetone/acetonitrile/methanol (1:1:1, v/v/v), containing 2.5 µM Metabolomics Amino Acid Mix Standard (Cambridge Isotope Laboratories), were added to 50 µl plasma or 100 µl brain homogenate (1 mg/ml protein content in water), respectively. After incubation (2 h, −20 °C) and centrifugation (20,000 × g, 10 min, 4 °C), supernatants were dried under vacuum and reconstituted in 25 µl or 50 µl methanol/acetonitrile (1:1, v/v) for LC-MS/MS analysis. Extracts were separated on a SeQuant ZIC-HILIC column (150 × 2.1 mm, 5 µm) using water with 5 mM ammonium acetate as eluent A and acetonitrile/eluent A (95:5, v/v) as eluent B, both containing 0.1% formic acid. The gradient elution was set as follows: isocratic step of 100% B for 3 min, 100% B to 60% B in 15 min, held for 5 min, returned to initial conditions in 5 min and held for 5 min. Flow rate was 0.5 ml/min. Full scan data acquisition with data-dependent MS$^2$ scans (top 10) was performed. Compound Discoverer 3.3 (Thermo Scientific) was used for data processing. Lactate was identified based on its exact mass ([M-H]$^-$ = 89.024), isotopic pattern, retention time (RT = 1.6 min) and fragmentation spectrum. We used an in-house library[79] as well as the online library mzCloud for comparison. For plasma samples, the area under the peak was normalized to the internal standard L-isoleucine

($^{13}C_6^{15}N$). For brain homogenate, normalization was performed based on the protein content, determined by a Bradford assay. An extraction blank as well as pooled samples at 4 concentrations were used as quality controls and for a quality control-based normalization.

## Behavioral testing
Mazes were cleaned with 70% ethanol solution between trials. Analysis of behavioral videos was conducted offline using tracking software (AnyMaze 4.98, Stoelting Europe, Ireland).

**Object place recognition test.** Cognitive function was tested using OPRT. Here, mice were adapted for five consecutive days, by placing them into a handling box ($20 \times 20$ cm) for 5 min. On day 6–8, mice were allowed to explore an open field ($35 \times 35$ cm, 19 lux) for 10 min each day, to habituate to the OPRT area. On day 9, mice were confronted with two objects placed at the edges of the open field area and allowed to explore them for 10 min. After a 1-h break in the home cage, mice were placed back in the open field, where one of the objects had been displaced, for 5 min and the exploration time for each object was determined. We calculated the preference index according to the following formula:

$$\text{preference index} = \frac{\text{Time}_{\text{displaced object}}}{\text{Time}_{\text{unaltered object}} + \text{Time}_{\text{replaced object}}} \quad (1)$$

**Barnes maze.** The Barnes maze experiment was conducted as described previously[80]. Briefly, the animals had four trials of maximally 3 min each per day on four consecutive days in the Barnes maze (Stoelting Europe, Ireland; technical specifications: diameter 91 cm, height 90 cm, 20 holes, hole diameter 5 cm; light intensity 360 lux at maze level) with an inter-trial interval of 15 min. On day 5 and day 16, probe trials took place. The escape box was replaced by a blind. The maze was rotated by 180° to prevent orientation on potentially present intra-maze cues. The animal was allowed to explore the maze for 90 s, thereafter, it was brought back into its home cage. Video analyses comprised semi-automatic measurements of latency to reach the target hole as well as the manual measurement of errors (head dips into non-target holes). As some mice tend to continue exploring the maze after reaching the target hole for the first time, we only report primary errors occurring before mice reached the target for the first time[81]. Furthermore, a search strategy was assigned to each trial according to the following criteria[81]. Direct search, the animal moved either directly to the target hole or to an adjacent hole before reaching the target for the first time; serial search, the animal visited at least two adjacent holes in a serial manner before reaching the target for the first time; mixed search, the animal showed an unorganized search pattern; no search, the animal visited less than 4 holes in 90 s.

**Elevated plus maze.** The elevated plus maze consisted of two closed and two open arms, each arm 5-cm wide, illumination 100 lux. The animal was placed at the intersection of the four arms and could explore the maze for 5 min. We measured the time spent in the closed arms.

**Y-maze.** To test spatial working memory, the animal was placed at the distal part of one arm of the maze (length of each arm: 20 cm, wall height: 20 cm, illumination: 30 lux at the center of the maze) and could explore the maze for 8 min. All arm entries (defined by entering with all four paws) were recorded and alternating triplets (consecutive entries into all three arms) were counted. Triplets could overlap. For example, if the arms are labeled arm A, B and C, the following sequence consists of two triplets: A-B-C-B-A. The alternation index is expressed as % and

calculated according to following formula:

$$\text{alternation index} = \frac{\text{Number of triplets} \times 100}{(\text{Total number of arm entries} - 2)} \quad (2)$$

## Middle cerebral artery occlusion
Ischemic stroke was induced by permanent occlusion of the distal middle cerebral artery in 8-week-old mice[82]. Pentobarbital was used for anesthesia (15 µl of 0.5% pentobarbital/g body weight, i.p.) and baseline EDTA plasma was collected from the tail vein. Then, mice were injected with glucose solution (50 mg in 200 µl normal saline, i.p.) or vehicle 10 min before MCAO. The surgery was performed under a microscope (Hund) and rectal temperature was maintained at 37 °C with a heating pad. First, a skin incision was made between the ear and the orbit on the left side and the temporal muscle was removed. Then, a burr hole was drilled and the exposed MCA was occluded by microbipolar electrocoagulation (Model ICC; Erbe). Finally, the skin incision was sutured and the mice were placed under a heating lamp. For sham surgery, mice were anesthetized and only a skin incision was made. Mice were deeply re-anesthetized with pentobarbital 30 or 50 min after occlusion, EDTA plasma was collected intracardially, and the ipsilateral and contralateral brain tissues were snap frozen in liquid nitrogen after perfusion with 15–20 ml Ringer's solution. For MS, brain tissue was homogenized in PBS or LC-MS grade water (Bullet blender STORM BBY 24 M, Next Advance) and protein content determined using Lowry's method or Bradford protein assay. If blood glucose levels did not rise to >11.1 mM 25 min after glucose administration, mice were excluded. The infarct size was determined 2 days after MCAO as described previously[50,51].

## Immunohistochemistry
Immunostainings for occludin, and 4-HNE were performed as reported previously with some modifications[83]. Sagittal cryosections (20 µm thick) from Ringer-perfused brains were methanol-fixed for 10 min at −20 °C. After blocking with 1% bovine serum albumin (BSA), slides were incubated overnight with rat anti-mouse CD31 antibodies (1:500, BD Pharmingen) together with rabbit anti-4-HNE (1:1,000, Calbiochem), or rabbit anti-mouse Ocln antibodies (1:1000, Proteintech Europe) at 4 °C. The next day, the slides were incubated with Alexa 488-labeled anti-rabbit (1:400, Invitrogen) and Cy3-labeled anti-rat (1:400, Jackson) containing DAPI (1 µg/ml) for 2 h at room temperature. Imaging was performed with a fluorescence microscope (DMI 6000 B, Leica Application Suite Advanced Fluorescence 2.5.0.6735) and analysis was done using ImageJ (2.0.0-rc-15/1.49 m). For analyzing oxidative stress, the CD31+4-HNE+ vessel length was measured. Occludin intensity was detected by creating a mask of the CD31+ staining, which restricted the detected signal to brain vessels.

GFAP staining was performed as described above, with the exception that sections were incubated with 4% paraformaldehyde (PFA) for 20 min at room temperature instead of the methanol fixation step. For Iba1 staining, mice were perfused with 15–20 ml Ringer's solution and 15–20 ml 4% PFA 3 h and 24 h after DMG injection, followed by post-fixation in 4% PFA (4 °C, 24 h). Then, brains were cryo-protected with 30% sucrose and snap frozen in isopentane. Cryosections (20 µm thick) were air-dried and rehydrated with PBS for 5 min. After antigen retrieval in citrate buffer (pH 6), immunostaining was performed as described before. Rabbit anti-mouse Iba1 (1:100, Wako) or rabbit anti-mouse GFAP (1:400, Dako) primary antibodies and Alexa 488-labeled donkey anti-rabbit (1:400, Invitrogen) secondary antibodies were used. GFAP intensity and number of Iba1+ cells were determined using ImageJ.

## Immunocytochemistry in primary brain endothelial cells
Primary brain endothelial cells (PBEC) were treated with 500 µM dimethylglyoxal or methylglyoxal in full PBEC medium (DMEM-F12,

20% plasma derived serum, penicillin/streptomycin (100 U/ml; 0.25 g/ml), 1% endothelial cell growth factor, 2 mM L-glutamine, 15 U/ml heparin) for 3 or 24 h. Then, PBEC were washed with PBS and fixed with 500 µl methanol for 10 min at −20 °C. After another PBS washing step, blocking was done by 5% BSA in PBS for 30 min at room temperature. Rabbit anti-mouse Ocln antibodies (1:1000, Proteintech Europe) in blocking solution were added and PBEC incubated overnight at 4 °C. The next day, the cells were incubated with Alexa 488-labeled donkey-anti-rabbit antibodies (1:400, Invitrogen) in blocking solution for 1 h at room temperature. Then, nuclei were stained with DAPI (1 µg/ml) and the occludin staining was analyzed immediately on a fluorescent microscope (DMI 6000B, Leica) and intensity determined with ImageJ software.

## Cell lines and isotopic labeling

The mouse brain endothelial cells bEnd.3 (ATCC, CRL-2299) and mouse hippocampal neuronal HT22 cells were cultured in DMEM containing 25 mM glucose (FG0445, Merck), 10% fetal calf serum (FCS), and penicillin/ streptomycin (100 U/ml; 0.1 mg/ml)) at 37 °C and 5% $CO_2$. For treating cells with $^{13}C$-labeled glucose or pyruvate, glucose-free DMEM (11966-025, gibco) with the same supplements was used. The medium was supplemented with 25 mM $[^{13}C_6]$-glucose (Cambridge Isotope Laboratory) or native $[^{12}C]$-glucose for 6, 24, or 48 h. For atmospheric control, the InvivO$_2$ sterile culture hood (Baker Ruskin) was employed. O$_2$ was set to 0.1–0.5% to obtain hypoxic conditions (37 °C, 5% $CO_2$). Medium was snap frozen in liquid nitrogen and stored at -80 °C until LC-MS measurement. For pyruvate-related isotopic labeling of dimethylglyoxal, we added 100 mM $[^{13}C_1$-1]-, $[^{13}C_1$-2]-, $[^{13}C_1$-3]-, $[^{13}C_3]$-pyruvate (Cambridge Isotope Laboratory), or $[^{12}C_3]$-pyruvate to glucose-free DMEM for 24 h (Fig. 5a) or 4 mM $[^{13}C_3]$-pyruvate for 6 h (Fig. 5c).

## Ilvbl overexpression

To overexpress Ilvbl, restriction sites for PacI and AscI restriction enzymes were added to the Ilvbl cDNA sequence from mouse primary brain endothelial cells (PBEC) at the 5′- and 3′-terminus, respectively, using Phusion® High-Fidelity DNA polymerase (BioLabs). Then, purified PCR products were inserted into the pFastBac-AAV plasmid containing a CAG promotor to drive Ilvbl expression. Correct integration of the fragment was verified by AhdI, SmaI, and PacI/AscI restriction (BioLabs) as well as sequencing (GATC Biotech AG).

The Ilvbl-expressing plasmid pFB-CAG-ILVBL was transfected into bEnd.3 cells using Lipofectamine 3000 reagent (Thermo Fisher) and cells were treated with $[^{13}C_6]$-glucose and hypoxia 24 h later. Controls received the plasmid pFB-CAG-GFP.

## Western blot

IgG or ILVBL protein levels were analyzed by Western blot. Here, after perfusion with Ringer's solution, brain homogenates containing 100 ng protein or bEnd.3 cell lysates were separated by sodium dodecyl sulfate polyacrylamide gel electrophoresis and transferred to nitrocellulose membranes. After blockage with 5% milk, the membranes were incubated with rabbit anti-mouse ILVBL (ALS, 1:400, Abcam), mouse anti-actin (1:2,000, Millipore), or horseradish peroxidase (HRP)-coupled goat anti-mouse IgG (1:5000, Jackson) antibodies at 4 °C overnight. For actin and ILVBL detection, the HRP-coupled goat anti-mouse (1:10,000, Jackson) or HRP-coupled goat anti-rabbit (1:10,000, Jackson) antibodies were added the next day and incubated at room temperature for 2 h. Chemiluminescence was induced using the Pierce™ SuperSignal West Femto Sensitivity Substrate (1:4, Thermo Scientific) and detected with the Fusion Solo S (17.01) detection system. ILVBL and IgG signals were normalized to actin. Uncropped scans of blots are provided in the Source Data file.

## Detection of reactive oxygen species

Oxidative stress in bone-marrow-derived macrophages (BMDM), bEnd.3 cells, astrocytes, and HT22 was determined using the ROS-sensitive fluorescent dye CM-H$_2$DCFDA (Invitrogen). A stock solution (4 mM in DMSO) was diluted 1:1000 in PBS and 200 µl of the 4 µM solution was added to cells growing on a 96-well plate. The dye was incorporated by the cells for 30-40 min at 37 °C and 5% $CO_2$. Then, dye solution was removed, cells were treated with PBS, dimethylglyoxal or methylglyoxal for 10 min, and the fluorescence signal was detected at 495 nm excitation with a 510-nm emission wavelength. As positive control, 10 µM hydrogen peroxide was added at the same time. Cell-free wells served as negative controls. For these experiments, HT22 cells were grown on poly-d-lysine (1 mg/ml, Sigma-Aldrich)- coated wells.

## RT real-time PCR after in-vitro dimethylglyoxal treatment

Confluent bEnd.3 or HT22 cells were treated with 100 µM or 500 µM dimethylglyoxal for 24 h in DMEM containing 25 mM glucose, 10% FCS, and penicillin/streptomycin (100 U/ml; 0.1 mg/ml). BMDM were treated with 100 µM or 500 µM DMG during a 48-h polarization phase in BMDM growth medium (DMEM (without glucose), 10% heat-inactivated FCS, penicillin/streptomycin (100 U/ml; 0.1 mg/ml), and 5 mM glucose)) with lipopolysaccharides (LPS, E. coli 0111:B4, 100 ng/ml; Sigma-Aldrich) or interleukin-4 (IL-4, 10 ng/ml; ReproTech) for M1 or M2 polarization, respectively.

RNA was isolated from bEnd.3, HT22 cells and BMDM using the NucleoSpin 96 RNA purification kit (Macherey-Nagel) according to the manufacturer's instructions. Then, 400 ng of RNA was transcribed to cDNA with the Cloned AMV First-Strand Synthesis kit (Life Technologies). Primers for real-time PCR are listed in Supplementary Table 3. Platinum SYBR Green qPCR SuperMix (Invitrogen) was used with 40 cycles of the following protocol: 2 min at 50 °C, 2 min at 95 °C, 15 s at 95 °C, and 1 min at 60 °C with the ROCHE Light cycler. Ppia was used as housekeeping gene for data normalization according to the ΔΔCt method.

## Preparation of primary cells

PBEC and BMDM were prepared from 8- to 10-week-old mice. The animals were anesthetized with isoflurane and killed by decapitation. PBEC were prepared from the cortex and BMDM from bone-marrow of femur and tibia as described previously[14,84]. Primary cortical astrocytes were prepared from postnatal mice at day 1 as described elsewhere[85].

## Statistical analysis

Data were analyzed using GraphPad Prism 8 (GraphPad Software) and SPSS 25 (IBM). Sample sizes were planned based on a power analysis by G*Power using the dimethylglyoxal levels as primary outcome parameter. Significance was considered when $p < 0.05$. Depending on the dataset and experimental design, different statistical methods were used as indicated in Supplementary Tables 5 and 6. Parametric statistics (e.g., t-Test, ANOVA) were only applied if assumptions were met, i.e., datasets were examined for Gaussian distribution by D'Agostino-Pearson test, aided by visual inspection of the data, and homogeneity of variances by Brown-Forsythe, Levene's or F-test (depending on the statistical method used). If assumptions for parametric procedures were not met or could not be reliably assumed due to small sample size, non-parametric methods were used as indicated. Two-tailed tests were applied if not indicated otherwise. Greenhouse-Geisser correction was used in ANOVA statistics if the sphericity assumption was violated (Mauchly test). Outliers were identified by the ROUT method ($Q = 1\%$). No data points were excluded if not stated otherwise in Supplementary Tables 5 and 6.

## Reporting summary

Further information on research design is available in the Nature Portfolio Reporting Summary linked to this article.

## Data availability

All data needed to recapitulate the results presented here can be found in the manuscript, figures and supplementary data or from the corresponding author on request. Source data are provided with this paper.

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

## Acknowledgements

We would like to thank Walter Raasch for help in measuring plasma cytokines. The research leading to these results received funding from the Deutsche Forschungsgemeinschaft (SCHW 416/7-1) and from the European Research Council (ERC) under the European Union's Horizon 2020 research and innovation program (grant agreement No 810331) to M.S., V.P., and R.N.

## Author contributions

S.R. and M.S. designed the study. R.C., S.R., T.G., T.C.F., Z.S., A.A.P., S.B., I.S., and V.D. performed in vitro and in vivo experiments and analyzed the data. R.C., J.I., S.R., A.O., and K.B. performed LC-MS/MS analyses. P.P.N., T.F., K.O., V.P., and R.N. provided reagents and conceptual input. S.M. and S.M.M. performed the clinical study. R.C., S.R., J.I., and M.S. wrote the manuscript. All authors reviewed and commented on the manuscript.

## Funding

## Competing interests

The authors declare no competing interests.

## Additional information

[1]Institute for Experimental and Clinical Pharmacology and Toxicology, University of Lübeck, Lübeck, Germany. [2]German Research Centre for Cardiovascular Research (DZHK), partner site Hamburg/Lübeck/Kiel, Lübeck, Germany. [3]Bioanalytic Core Facility, Center for Brain Behavior and Metabolism, University of Lübeck, Lübeck, Germany. [4]Department of Physiology, CIMUS, University of Santiago de Compostela-Instituto de Investigación Sanitaria, Santiago de, Compostela, Spain. [5]Department of Medicine I and Clinical Chemistry, University Hospital Heidelberg, Heidelberg, Germany. [6]German Center for Diabetes Research (DZD), Munich, Neuherberg, Germany. [7]Neurogenetic Systems Analysis Group, Institute of Neurogenomics, Helmholtz, Munich, Neuherberg, Germany. [8]Institute of Human Genetics, School of Medicine, Technical University of Munich, Munich, Germany. [9]Univ. Lille, Inserm, CHU Lille, Laboratory of Development and Plasticity of the Neuroendocrine Brain, Lille Neuroscience & Cognition, UMR-S 1172, DISTALZ, EGID, Lille, France. [10]Institute for Endocrinology and Diabetes, University of Lübeck, Lübeck, Germany. [11]Department of Medicine I, University Hospital Schleswig-Holstein Campus Lübeck, Lübeck, Germany. [12]Present address: Functional Genomics Center Zurich, ETH Zurich, Zurich, Switzerland. [13]These authors contributed equally: Sina Rhein, Riccardo Costalunga, Julica Inderhees. ✉e-mail: markus.schwaninger@uni-luebeck.de

