## [Peer Review File · Nature Communications]

The reactive pyruvate metabolite dimethylglyoxal mediates neurological consequences of diabetesREVIEWER COMMENTS

Reviewer #1 (Remarks to the Author):

In this manuscript, Rhein, Costalunga and colleagues describe a novel pathway of diabetes related complications, namely the production of dimethylglyoxal (DMG) from lactate-derived pyruvate. The results highlight a novel pathway that is “distinct from other complications”, and thus the findings are quite interesting. The metabolic tracing experiments using ^{13}C are comprehensive, cleverly designed, and informative, and the inclusion of human data was also a major strength of the study. Despite these strengths, the study raised some concerns in both its choice of model systems, output measures, and most importantly data interpretation. For example, while the synthesis of in vitro overexpression studies and in vivo knockout (Ilvbl) were another strength, the findings in the stroke model were not overwhelming. More importantly it was not at all clear the relevance of the MCAO model, nor why the authors chose this rather than a diabetes model. Additionally, the brain-based measures shown, including the in vitro studies, come across as a bit disjointed. There are also major concerns with the reproducibility of the elevated DMG phenotype in the models employed. Finally, while the manuscript was generally well written and organized, there were several instances in which the results did not appear to match the interpretation and its implications in regards to human disease. More specific comments are laid out below:

Major comments:

The logic laid out in the abstract is quite confusing. It is not at all clear how the usage of lactate by neurons or endothelial cells justifies a distinct cause of diabetes-related cognitive impairment. Similar to the abstract, the first half of the introduction is very confusing in the arguments it makes and how it justifies the current study. How is acute ischemic stroke (a model the authors employ later in the manuscript) related to this? More importantly, hardly any of the strong statements made in the first third of the introduction are backed by references.

While the connection between diabetes and dementia risk is well established, there is

actually some disagreement on whether this relationship also extends to cognitive impairment. Further, the effects appear to involve specific cognitive domains and do not necessarily equate between T1D and T2D. While the authors show data from T2D human plasma, the STZ model of T1D was the only mouse model studied here; additionally, the STZ mice were untreated, and have plasma [glucose] higher than their human T1D counterparts. Confirmation of these findings in a model of T2D would seem to be a critical missing piece here.

Another focus of the manuscript, and its relevance is highlighted numerous times in the abstract and introduction, is the topic of diabetes-related cognitive impairment. While the authors stress this feature, their results only address this in Fig 9d where they use the OPRT. This is not the most robust cognitive test and no other alternative behavioral or cognitive measures are described. In essence, results seem to be over interpreted should they match the focus of the abstract, introduction and even the “neurological complications” title in this regard.

The statement on line 135-6 does not seem to appropriately reflect the data provided.. The data from human patients, while very important, is only from plasma, not brain. These data on human diabetic serum DMG levels are also less noticeably altered than 3-DG and at least in the T2D patients, similar in scope to Gx and MG. In the STZ mice, DMG levels in plasma seem to be on par, or less affected, than 3-DG. Additionally, the increases in DMG in brain seem to be of similar degree to the increase in brain MG.

In a related question, what are the lactate and pyruvate levels of these mice in plasma and brain?

Why were 3-DG and DMG levels not also increased in the chow control STZ animals in Fig 3 but were in the cohort in Fig 1. Similarly DMG was up at 12 weeks in Fig 1 but not at 14 weeks in Fig 9. How consistent is this feature? If it is not a clearly replicable feature of the STZ model, then similar to the comments above, this may not be the best model to study the DMG pathway or to reflect the human condition. These discrepancies raise concerns about the reproducibility of the phenotype across the study, and therefore the overall interpretation.

The statement of line 165 (“the most prominent effects”) may not be entirely accurate. If assessed by % then yes, but it appears as though the “Rel. Area” numbers for MG increase ~15 while for DMG that number is closer to ~8. Which is the most appropriate way to assess these changes, by change in concentration values?

The selection of the bEnd.3 cell line makes sense given the earlier statement of neurons and endothelial cells utilizing lactate. Why then was a neuronal line not included in the studies in Fig 8? What is the justification for utilizing the BMDMs?

Minor comments:

If DMG arises from lactate-derived pyruvate, and DMG is higher in diabetes, are [lactate] concentrations also higher in diabetes compared to normal?

In Fig 4a and 4b, what were dicarbonyl levels within the cells themselves?

What were the stats on Fig 4b-c. Shouldn't repeated measures ANOVA used for a time course? No significance is noted with a symbol on these figures.

Is the difference in DMG between baseline and 50 min vehicle not significant? It appears to be elevated. Even if not $p < 0.05$, there is a clear trend of higher DMG in vehicle. How do the authors explain this?

Is DMG increased in the brain because of transport or because of differences in endothelial/neuronal usage? Astrocyte or other cell type production?

Reviewer #2 (Remarks to the Author):

Dear Editor,

The manuscript entitled “A reactive pyruvate metabolite mediating neurological

complications of diabetes” is an original manuscript describing the formation of dimethylglyoxal from pyruvate and its effects in causing neuronal damage in diabetes. The role of dicarbonyls in the development of diabetic complications has been described by numerous articles in the last few years. Here, the authors describe a new metabolite that may be particularly important for neuronal damage and cognitive impairment in diabetes. The manuscript is very well constructed with relevant data for the field. However, after reading it, I have some questions.

1. The authors claim in the introduction and abstract that neurons consume mostly lactate, which originates much fewer dicarbonyls, suggesting the existence of another pathway. However, in the animal model described in Figure 1, the brain methylglyoxal levels are elevated in diabetic rats, even before the same happens in the blood. How do authors explain such results?
2. The authors mention that DMG is mostly formed in the brain due to the higher pyruvate utilization. However, in the animal models studied, they also observe a faster increase in plasma DMG levels than dicarbonyls like MG and Gx. What is the expected peripheral DMG source and how do the plasma levels so quickly change if its source is mainly the brain? The same is true for the results presented in Figure 4. If ischemia is induced only in the brain, how do DMG plasma levels increase?
3. One of my main concerns is the physiological significance of DMG doses used for i.p. injection. 100 mg/Kg is apparently a very high dosage, which is difficult to observe in vivo in physiological conditions. The authors claim that Ilvbl participates in the formation of DMG. If we assume that DMG is important for the development of diabetic complications and Ilvbl-positive mice have higher DMG levels, these mice should develop diabetes-like complications. If they don't, we may assume that DMG is not so relevant for the development of such complications, and the alterations observed in DMG-treated mice were only observed due to the high dose used. Since HBL was found to be increased in Ilvbl-positive mice, it would be interesting to compare such levels with those found in DMG-treated mice. This would confirm the physiological significance of the dose used.
4. Similar concerns may be raised in Fig 8, regarding the effects of DMG in inducing ROS in cultured cells. Oxidative stress was observed mainly after exposure to 500uM DMG. This is an extremely high dose. For MG, ~200 uM is usually the EC50 for cell viability. In ex vivo environments (not using immortalized cell lines), the EC50 was shown to be around 100 uM.

Authors here show that 100 μ M DMG does not induce major oxidative stress changes. The same is true for the neuroinflammation experiments described in Fig 8. All the changes were achieved only after 500 μ M exposure.

5. Neuroinflammation experiments were also performed in DMG-treated mice. Why not perform them in Ilvbl-positive mice?

6. The physiological significance of the experiments is also highlighted by the use of the same dosage for i.p. and oral DMG administration. The authors used 100 mg/kg in both protocols, although they are expected to produce completely different DMG bioavailability. This raises doubts about the physiological meaning of especially the i.p. protocol.

Interestingly, in both figures 7 and 9 authors show plasma concentrations in similar ranges (2-5 nM) with such completely different protocols, which is not expected. Authors should address this question and confirm whether this is real or an experimental error.

7. Since authors show the rapid disappearance of DMG from the plasma and suggest a transformation into HBL, what is the evidence about DMG intestinal absorption? Authors show increased DMG plasma levels after 6 weeks of oral administration, but not after 14 weeks. On the other hand, increased HBL levels were shown. Is there evidence of DMG absorption both in the free form and in the BHL form?

Reviewer #3 (Remarks to the Author):

The manuscript by Markus Schwaninger et al. details the discovery of dimethylglyoxal as a neuro-damaging substance derived from pyruvate, which leads to cognitive dysfunction in diabetes. Although the discovery seems serendipitous, the authors meticulously construct a comprehensive series of experiments to trace the origin and destiny of dimethylglyoxal. The authors have identified α -dicarbonyl dimethylglyoxal as a product of cellular pyruvate metabolism. Although dimethylglyoxal has previously been recognized as a microbial metabolite and flavor compound, its biological activity and origin in mammals remained unclear. Unlike other α -dicarbonyls formed as by-products of glycolysis, dimethylglyoxal was shown to originate from pyruvate metabolism. This process involves the condensation of C2 and C3 atoms from two pyruvate molecules after decarboxylation. Interestingly, the gene *Ilvbl*, homologous to yeast acetolactate synthase, contributes to dimethylglyoxal formation in animals. Elevated dimethylglyoxal levels were detected in the blood and brain tissues of

experimental and clinical diabetes cases. It was found to impair blood-brain barrier function and cause cognitive defects in mice similar to those seen in diabetes-induced cognitive impairment. Dimethylglyoxal's synthesis pathway could offer new insights for addressing carbonyl stress. Combining metformin with insulin proved more successful in reducing dimethylglyoxal levels than insulin alone in diabetes patients. This discovery could potentially enhance metformin's efficacy in preventing diabetes complications. Targeting ILVBL, a druggable enzyme, might open up novel treatment avenues for diabetic complications.

The experimental work is nicely developed including an animal model of diabetes, humans, and cell cultures to confront the different hypotheses. The questions that appear while reading the experiment have been approached in the next step and support the hypothesis. In those aspects within my expertise, the materials and methods section has been adequately described.

My only comment is that probably the authors could risk a bit more in the title including the name of the compound. Saying that I think the manuscript deserves to be published in Nature Communications.

Response to editor and reviewers

We would like to thank the reviewers for their helpful comments on the manuscript that have guided us during the thorough revision of the manuscript. Following the comments, we have performed extensive new experiments. To include the new data, the following figure panels were added or changed during the revision:

- Fig. 1g (db/db mice),
- Fig. 4b (MG production during hypoxia),
- Fig. 4c (DMG production during hypoxia),
- Fig. 8c (ROS production in HT22 cells),
- Fig. 8g (DMG effect on gene expression in HT22 cells),
- Fig. 9a (Intestinal absorption of [¹³C₄]-DMG),
- Fig. 9d (DMG plasma levels after 14 weeks of DMG treatment),
- Supplementary Fig. 1c (DMG plasma levels and additional parameters in 2nd cohort of STZ-treated mice),
- Supplementary Fig. 1d (DMG brain levels and additional parameters in 2nd cohort of STZ-treated mice),
- Supplementary Fig. 1e, f (body weight over time and blood glucose of db/db mice),
- Supplementary Fig. 1g (DMG levels and body weight in HFD-fed mice),
- Supplementary Fig. 2a, b (3-DG and Gx production during hypoxia),
- Supplementary Fig. 4b (concentration response curve of DMG on ROS production),
- Supplementary Fig. 5a (Chromatogram and fragmentation spectrum of derivatized ¹³C₄-labeled DMG)
- Supplementary Fig. 5b, c (MG plasma and DMG brain levels after 14 weeks of DMG treatment)
- Supplementary Fig. 5d-g, k, l (Y-maze, open field, elevated plus maze and Barnes maze of STZ and DMG-treated mice)
- Supplementary Fig. 5h, i (Correlation between DMG concentrations in brain and central-zone speed or time immobile).

All changes in the text are marked in **red font**.

During the revision we discovered a mistake in the calculation of dimethylglyoxal concentrations. A factor of 10 was missing in the original Excel sheet and this error was carried over into subsequent calculations. Now all scientists involved in the LC-MS measurements (SR, RC, JI, TG) have reviewed the calculation once again and agree that it is correct. We apologize for this mistake but luckily all values were affected by the same factor and significance levels were not changed.

Reviewer #1 (Remarks to the Author):

In this manuscript, Rhein, Costalunga and colleagues describe a novel pathway of diabetes related complications, namely the production of dimethylglyoxal (DMG) from lactate-derived pyruvate. The results highlight a novel pathway that is “distinct from other complications”, and thus the findings are quite interesting. The metabolic tracing experiments using ¹³C are comprehensive, cleverly

designed, and informative, and the inclusion of human data was also a major strength of the study. Despite these strengths, the study raised some concerns in both its choice of model systems, output measures, and most importantly data interpretation. For example, while the synthesis of *in vitro* overexpression studies and *in vivo* knockout (Ilvbl) were another strength, the findings in the stroke model were not overwhelming. More importantly it was not at all clear the relevance of the MCAO model, nor why the authors chose this rather than a diabetes model. Additionally, the brain-based measures shown, including the *in vitro* studies, come across as a bit disjointed. There are also major concerns with the reproducibility of the elevated DMG phenotype in the models employed. Finally, while the manuscript was generally well written and organized, there were several instances in which the results did not appear to match the interpretation and its implications in regards to human disease. More specific comments are laid out below:

Response: We are glad that the reviewer sees strengths in our study. Concerning the models, we apologize that we did not justify their selection more carefully. In addition to two diabetes models (STZ and db/db mice, the latter being added during the revision), we have combined MCAO and hyperglycemia in a model of hyperglycemic stroke. This condition is of high clinical relevance because hyperglycemia is common in acute stroke patients and is known to aggravate ischemic brain damage¹. The co-occurrence of hyperglycemia and hypoxia/ischemia allowed us to investigate the formation of dimethylglyoxal in a clinically important disease *in vivo*. These considerations are included in the revised manuscript (page 8, lines 25-27).

With respect to the reproducibility of dimethylglyoxal production, we are very convinced of the solidity of the data because elevated dimethylglyoxal levels were found in four independent cohorts of mice treated with STZ (Fig. 1, Fig. 3, Supplementary Fig. 1, Fig. 6e) as well as in db/db mice (added during the revision) and in human diabetes patients.

Regarding the human data, we have adapted the interpretation following the reviewer's comments.

Major comments:

Reviewer #1: "The logic laid out in the abstract is quite confusing. It is not at all clear how the usage of lactate by neurons or endothelial cells justifies a distinct cause of diabetes-related cognitive impairment. Similar to the abstract, the first half of the introduction is very confusing in the arguments it makes and how it justifies the current study. How is acute ischemic stroke (a model the authors employ later in the manuscript) related to this? More importantly, hardly any of the strong statements made in the first third of the introduction are backed by references."

Response: We are glad about this feedback and have improved the argumentation line in the abstract and introduction (page 3; page 4, lines 6-9, 12). Our main points can be summarized as follows. Classical α -dicarbonyls such as methylglyoxal are formed in the first phase of glucose metabolism to lactate. In the brain, the first phase is outsourced to glial cells while neurons, and partly endothelial cells, take up lactate. Therefore, methylglyoxal production mainly occurs in glial cells and not in neurons. The discovery that dimethylglyoxal is formed from lactate as a toxic by-product suggests a mechanism of how lactate metabolism can pose a metabolic risk for neurons and endothelial cells. Following the reviewer's comment, we have cited several additional references in the introduction.

Reviewer #1: "While the connection between diabetes and dementia risk is well established, there is actually some disagreement on whether this relationship also extends to cognitive impairment. Further, the effects appear to involve specific cognitive domains and do not necessarily equate

between T1D and T2D. While the authors show data from T2D human plasma, the STZ model of T1D was the only mouse model studied here; additionally, the STZ mice were untreated, and have plasma [glucose] higher than their human T1D counterparts. Confirmation of these findings in a model of T2D would seem to be a critical missing piece here.”

Response: Several published studies link cognitive deficits and experimental type-2 diabetes^{2, 3, 4, 5, 6}. In this context, the reviewer is right to raise the question of whether type-2 diabetes mice also show elevated dimethylglyoxal plasma levels. In the well-established db/db mouse model of type-2 diabetes, we found markedly elevated dimethylglyoxal plasma concentrations (Fig. 1g). As the db/db mice were obese and hyperglycemic, we also investigated a cohort of mice with diet-induced obesity but no overt diabetes. Mice with diet-induced obesity had unaltered dimethylglyoxal plasma concentrations (Supplementary Fig. 1g), suggesting that type-2 diabetes and not obesity leads to increased dimethylglyoxal plasma levels in db/db mice and likely to the known cognitive impairment.

Reviewer #1: “Another focus of the manuscript, and its relevance is highlighted numerous times in the abstract and introduction, is the topic of diabetes-related cognitive impairment. While the authors stress this feature, their results only address this in Fig 9d where they use the OPRT. This is not the most robust cognitive test and no other alternative behavioral or cognitive measures are described. In essence, results seem to be over interpreted should they match the focus of the abstract, introduction and even the “neurological complications” title in this regard.

Response: We appreciate the cautionary notes given by the reviewer in this and the previous comment about the association between diabetes and cognitive impairment. We acknowledge that we might have overemphasized this link in the original manuscript and have toned down statements about cognitive impairment and neurological complications of diabetes (page 15, lines 12-14). Following the reviewer’s suggestion, we have extended the behavioral analysis of STZ- and dimethylglyoxal-treated mice, performing object place recognition test, open field, Y-maze, elevated plus maze, and Barnes maze. The data demonstrate subtle but definite deficits (Fig. 9, Supplementary Fig. 5). In addition to impaired memory in the object place recognition test, we found that dimethylglyoxal concentrations in brain correlated with the time mice were immobile in the open field and inversely correlated with the speed in the inner zone (Supplementary Fig. 5h, i). Compared to models of severe dementia, cognition in the STZ- and dimethylglyoxal-treated mice was only slightly impaired, which corresponds very well to the clinical findings in diabetes patients. We have revised the text and explicitly state that experimental diabetes is associated with mild cognitive deficits (page 15, lines 12-14).

Reviewer #1: “The statement on line 135-6 does not seem to appropriately reflect the data provided.. The data from human patients, while very important, is only from plasma, not brain. These data on human diabetic serum DMG levels are also less noticeably altered than 3-DG and at least in the T2D patients, similar in scope to Gx and MG. In the STZ mice, DMG levels in plasma seem to be on par, or less affected, than 3-DG. Additionally, the increases in DMG in brain seem to be of similar degree to the increase in brain MG.”

In a related question, what are the lactate and pyruvate levels of these mice in plasma and brain?

Response: The reviewer objects to the statement in the original manuscript that in human and experimental diabetes dimethylglyoxal rises more prominently than glycolysis-dependent dicarbonyls, especially in brain. We maintain the view that in most samples from diabetes patients or mice with experimental diabetes, dimethylglyoxal levels increased more prominently than the other α -dicarbonyls and this was also true in the now added db/db mouse model (fold-change of DMG, 1.96; MG, 1.89; Gx, 1.06; 3-DG, 1.72, Fig. 1g). However, as the reviewer points out, there are some

exceptions and we agree that the original statement was an abbreviated summary. Therefore, we have specified the sentence as follows: “Overall, these data indicate, that in human and experimental diabetes dimethylglyoxal increases significantly and often more than glycolysis-dependent dicarbonyls, with a marked increase in the brain” (page 7, lines 20-22).

Previous publications in the literature have reported higher lactate concentrations in blood of diabetes patients^{7,8} as well as the hippocampus of STZ-treated rats¹⁰. In contrast, lower pyruvate concentrations in blood and hippocampus have been reported in the STZ model^{10,11,12}. To answer the related question of the reviewer with our experiments, we have adapted the LC-MS method to detect pyruvate in addition to dicarbonyls and have measured lactate with another LC-MS technique (page 21, lines 1-2; page 22, lines 16-17). In the STZ model we found slightly decreased plasma levels of pyruvate and an increased ratio of lactate to pyruvate in plasma (Supplementary Fig. 1c), probably reflecting the lower NAD⁺/NADH ratio that has been reported in diabetes by several groups¹³. The slightly lower pyruvate levels in plasma do not explain the increased dimethylglyoxal concentrations, but other compartments than plasma might be more relevant for dimethylglyoxal production. These considerations have been included in the revised manuscript (page 10, lines 4-14).

Reviewer #1: “Why were 3-DG and DMG levels not also increased in the chow control STZ animals in Fig 3 but were in the cohort in Fig 1. Similarly DMG was up at 12 weeks in Fig 1 but not at 14 weeks in Fig 9. How consistent is this feature? If it is not a clearly replicable feature of the STZ model, then similar to the comments above, this may not be the best model to study the DMG pathway or to reflect the human condition. These discrepancies raise concerns about the reproducibility of the phenotype across the study, and therefore the overall interpretation.

Response: In the original Fig. 3 we had only included the significant results when comparing the diets. In response to the reviewer’s question, we have now also included other p-values. In chow-fed mice, treatment with STZ led to a significant increase in 3-DG and an almost significant increase in dimethylglyoxal ($p = 0.058$, Fig. 3d). Four experimental cohorts of mice treated with STZ (Fig. 1, Fig. 3, Fig. 6e, Supplementary Fig. 1), db/db animals (Fig. 1) and the clinical study (Fig. 2) yielded highly consistent results and firmly support the conclusion that diabetes increases dimethylglyoxal levels.

The situation is different in the experiments described in Fig. 9, in which we treated mice with dimethylglyoxal in drinking water. In these experiments, we deliberately chose a moderate dimethylglyoxal dose ($100 \text{ mg kg}^{-1} \text{ day}^{-1}$) that should not lead to toxic levels. We performed the experiment in two cohorts of mice. In the first cohort (included in the original manuscript) dimethylglyoxal plasma concentrations were significantly elevated with 6 weeks of treatment but not after 14 weeks. During the revision of the manuscript, we repeated the experiment with the same dimethylglyoxal dose. In the revised manuscript we combined both cohorts of mice and found significantly elevated dimethylglyoxal concentrations also after 14 weeks. The slight increase in dimethylglyoxal concentrations 14 weeks on dimethylglyoxal treatment reflects the moderate dose in this approach. In contrast, the dimethylglyoxal increase in experimental and clinical diabetes was highly reproducible.

Reviewer #1: “The statement of line 165 (“the most prominent effects”) may not be entirely accurate. If assessed by % then yes, but it appears as though the “Rel. Area” numbers for MG increase ~15 while for DMG that number is closer to ~8. Which is the most appropriate way to assess these changes, by change in concentration values?

Response: We thank the reviewer for this valid comment. The time course study in Fig. 4b and c was performed with a limited number of samples per time point (< 5 samples per time point). Therefore,

statistical testing by repeated-measures ANOVA may not be appropriate (see also comment below). We used the time course for orientation and repeated the experiment with 6-hour hypoxia and 8 samples per group (Fig. 4b, c, Supplementary Fig. 2a, b). In the new dataset, we have now expressed dicarbonyl levels relative to the parallel 6-h normoxia group and not to the 30-min normoxia group as in the original figure. With this form of presentation, the proportional increases in methylglyoxal and dimethylglyoxal are quite similar (about 4-fold). Therefore, we have reworded the statement to: “As expected, hypoxia increased the production of [¹³C₃]-methylglyoxal and also had a pronounced effect on uniformly labeled dimethylglyoxal” (page 8, lines 23-25).

Reviewer #1: “The selection of the bEnd.3 cell line makes sense given the earlier statement of neurons and endothelial cells utilizing lactate. Why then was a neuronal line not included in the studies in Fig 8? What is the justification for utilizing the BMDMs?”

Response: We agree with the reviewer that a neuronal cell line is more relevant in the context of neurological complications of diabetes. Therefore, we performed experiments on the mouse hippocampal neuronal cell line HT22. The data confirm previous findings that dimethylglyoxal induces ROS formation and expression of pro-inflammatory genes (Fig. 8c, g).

We have chosen BMDMs because they mediate hyperglycemic ischemic brain damage associated with elevated α -dicarbonyl levels¹⁴ (Fig. 6). Moreover, in BMDMs pro- and anti-inflammatory genes can be investigated depending on the polarization of the cells. In the revised manuscript, we have outlined the reasons why we have used BMDMs (page 13, lines 7-8) but shifted the data into the supplement in view of the lower relevance to the story (Supplementary Fig. 4c, d, f, g).

Reviewer #1: “Minor comments:

If DMG arises from lactate-derived pyruvate, and DMG is higher in diabetes, are [lactate] concentrations also higher in diabetes compared to normal?”

Response: We thank the reviewer for this interesting question. As mentioned above, previous publications in the literature have reported higher lactate concentrations in blood of diabetes patients^{7,8} and the hippocampus of STZ-treated rats¹⁰. In contrast, lower pyruvate concentrations in blood and hippocampus have been reported in the STZ model^{10,11,12}. We have adapted the LC-MS method to detect pyruvate in addition to dicarbonyls and have measured lactate with another LC-MS technique (page 21, lines 1-2; page 22, lines 16-17). In the STZ model we found slightly decreased plasma levels of pyruvate and an increased ratio of lactate to pyruvate in plasma (Supplementary Fig. 1c), probably reflecting the lower NAD⁺/NADH ratio that has been reported in diabetes by several groups¹³. The slightly lower pyruvate levels in plasma do not explain the increased dimethylglyoxal concentrations, but other compartments than plasma might be more relevant for dimethylglyoxal production. These considerations have been included in the revised manuscript (page 10, lines 4-14).

Reviewer #1: “In Fig 4a and 4b, what were dicarbonyl levels within the cells themselves?”

Response: We thank the reviewer for this question. In response, we have measured labelled dicarbonyls in extracts of bEnd.3 cells cultured with [¹³C₆]-glucose. However, with the extraction method used, we could not detect labelled dicarbonyls in tissue extracts. Apparently, the extraction and measurement are not sensitive enough. This observation is mentioned in the revised manuscript (page 8, line 18-19).

Reviewer #1: “What were the stats on Fig 4b-c. Shouldn’t repeated measures ANOVA used for a time course? No significance is noted with a symbol on these figures.”

Response: In the time course experiments (original Fig. 4b and 4c), we included only a small number of samples (< 5 samples per group). Therefore, we did not perform statistical analysis. In response to the reviewer’s question, we have added the results of new experiments investigating the peak of dicarbonyl production after 6 h of hypoxia in 8 samples per group. Moreover, further experiments reproduced the effect of hypoxia on dimethylglyoxal production (Fig. 5c, Fig. 5g, Supplementary Fig. 2n).

Reviewer #1: “Is the difference in DMG between baseline and 50 min vehicle not significant? It appears to be elevated. Even if not $p < 0.05$, there is a clear trend of higher DMG in vehicle. How do the authors explain this?”

Response: The reviewer is right that the dimethylglyoxal plasma levels at baseline and 50 min differed significantly in the vehicle group (Fig. 4e, see Supplementary Table V). In this model of cerebral ischemia, basal glucose levels (about 6 mmol/l¹⁴) are apparently sufficient to lead to elevated dimethylglyoxal production. Of note, the increase was more pronounced in hyperglycemia (Fig. 4e). We mention this in the revised manuscript (page 9, lines 1-3).

Reviewer #1: “Is DMG increased in the brain because of transport or because of differences in endothelial/neuronal usage? Astrocyte or other cell type production?”

Response: We thank the reviewer for this interesting question. As dimethylglyoxal is a highly reactive compound, it seems questionable that it could be transported freely in the body. In favor of a local production by lactate-consuming cells such as endothelial cells and neurons, we found elevated brain concentrations in the ischemic brain (Fig. 4f, Fig. 6b). Moreover, the observation that *Ilvbl*^{+/+} and *Ilvbl*^{-/-} genotypes differed in dimethylglyoxal concentrations in brain but not in plasma (Fig. 6b, c; 6e, f) suggests a local dimethylglyoxal production. Nevertheless, we also found some evidence for the transport of dimethylglyoxal. After oral administration, labelled dimethylglyoxal was resorbed into the blood stream and could be detected in the brain (Fig. 9a), possibly explaining why we observed cerebral effects of orally administered dimethylglyoxal. These considerations are addressed in the discussion of the revised manuscript (page 17, lines 3-5).

Reviewer #2: “(Remarks to the Author):

Dear Editor,

The manuscript entitled “A reactive pyruvate metabolite mediating neurological complications of diabetes” is an original manuscript describing the formation of dimethylglyoxal from pyruvate and its effects in causing neuronal damage in diabetes. The role of dicarbonyls in the development of diabetic complications has been described by numerous articles in the last few years. Here, the

authors describe a new metabolite that may be particularly important for neuronal damage and cognitive impairment in diabetes. The manuscript is very well constructed with relevant data for the field. However, after reading it, I have some questions.”

Response: We thank the reviewer for the positive evaluation and are confident that we can answer the questions.

Reviewer #2 “1. The authors claim in the introduction and abstract that neurons consume mostly lactate, which originates much fewer dicarbonyls, suggesting the existence of another pathway. However, in the animal model described in Figure 1, the brain methylglyoxal levels are elevated in diabetic rats, even before the same happens in the blood. How do authors explain such results?”

Response: We appreciate this comment. The reviewer is right that in addition to dimethylglyoxal also the glycolysis-derived methylglyoxal increased in brain (Fig. 1f). In the brain, astrocytes perform glycolysis and produce methylglyoxal¹⁵. Whole tissue extracts of the brain represent astrocytic glycolysis plus lactate consumption by neurons and endothelial cells. In the revised manuscript, we have been careful not to equate brain and neurons or endothelial cells and to emphasize that brain extracts reflect diverse cell types (page 4, line 24; page 6, lines 20-21).

Reviewer #2 “2. The authors mention that DMG is mostly formed in the brain due to the higher pyruvate utilization. However, in the animal models studied, they also observe a faster increase in plasma DMG levels than dicarbonyls like MG and Gx. What is the expected peripheral DMG source and how do the plasma levels so quickly change if its source is mainly the brain? The same is true for the results presented in Figure 4. If ischemia is induced only in the brain, how do DMG plasma levels increase?”

Response: Based on our finding that dimethylglyoxal is a pyruvate metabolite, we take lactate/pyruvate metabolism in the tricarboxylic acid cycle as a proxy for dimethylglyoxal production. Although the brain is a major lactate consumer, the reviewer is right that it is not the only one. Lactate also fuels the tricarboxylic acid cycle in peripheral tissues, including muscle and adipose tissue¹⁶. In the revised manuscript we explicitly state that under normal conditions peripheral tissues oxidize lactate and likely produce dimethylglyoxal (page 10, lines 7-9). In stroke, ischemia increases lactate and dimethylglyoxal production in the brain (Fig. 4f, Fig. 6b). We consider it likely that dimethylglyoxal plasma levels rise partially as a spill-over from the brain. In addition, stroke leads to a rise of plasma glucose levels¹⁷ that may result in peripheral dimethylglyoxal production and further increase dimethylglyoxal plasma concentrations.

Reviewer #2 “3. One of my main concerns is the physiological significance of DMG doses used for i.p. injection. 100 mg/Kg is apparently a very high dosage, which is difficult to observe in vivo in physiological conditions. The authors claim that Ilvbl participates in the formation of DMG. If we assume that DMG is important for the development of diabetic complications and Ilvbl-positive mice have higher DMG levels, these mice should develop diabetes-like complications. If they don’t, we may assume that DMG is not so relevant for the development of such complications, and the alterations observed in DMG-treated mice were only observed due to the high dose used. Since HBL was found to be increased in Ilvbl-positive mice, it would be interesting to compare such levels with those found in DMG-treated mice. This would confirm the physiological significance of the dose used.”

Response: We appreciate the important question of whether exogenous dimethylglyoxal administration can mimic its endogenous production. We acknowledge that this is probably only

partially the case and explicitly point this out in the revised manuscript (page 17, lines 13-15). Being a highly reactive compound, dimethylglyoxal is likely to exert effects at the site of production and to travel only to a limited degree in the body. Nevertheless, during the revision of the manuscript we could prove that after oral administration labelled dimethylglyoxal is absorbed to some degree and even detected in the brain of mice (Fig. 9a). Regarding the concern that the administered dose might be too high, please note that we used two different dosing regimens, a single i.p. injection in Fig. 7 and the prolonged administration of dimethylglyoxal in the drinking water in Fig. 9, which happen to use the same per-weight dose (100 mg/kg body weight) but significantly different administration times (single injection vs chronic administration). The i.p. dose was deliberately high to search for dimethylglyoxal adducts in the brain and dimethylglyoxal plasma concentrations were clearly higher than the physiological concentrations we measured under basal conditions (Fig. 7). In contrast, the oral dose administered only slightly elevated dimethylglyoxal plasma concentrations at 6 and 14 weeks to a range (6 weeks, 24 nM; 14 weeks, 33 nM, Fig. 9) that was not higher than the levels observed in diabetes (68 nM, Fig. 1g) and even did not increase brain concentrations of dimethylglyoxal (Supplementary Fig. 5c). We are more concerned that the chronically administered dose might have been too low to completely mimic the detrimental effects of the body's own production in diabetes.

Please note that we have used *Ilvbl*-knockout mice. The *Ilvbl*-positive animals are genetically wild-type littermates of the *Ilvbl*^{-/-} mice derived from heterozygous crosses. As wild-type animals, *Ilvbl*-positive mice have been investigated (Fig. 9) and they developed diabetic complications in the form of behavioral alterations.

Reviewer #2 “4. Similar concerns may be raised in Fig 8, regarding the effects of DMG in inducing ROS in cultured cells. Oxidative stress was observed mainly after exposure to 500uM DMG. This is an extremely high dose. For MG, ~200 uM is usually the EC50 for cell viability. In ex vivo environments (not using immortalized cell lines), the EC50 was shown to be around 100 uM. Authors here show that 100 uM DMG does not induce major oxidative stress changes. The same is true for the neuroinflammation experiments described in Fig 8. All the changes were achieved only after 500 uM exposure.”

Response: In response to the reviewer's comment, we have performed additional experiments testing the effect of lower dimethylglyoxal concentrations on ROS production. Importantly, dimethylglyoxal stimulated ROS production even in a concentration of 20 μ M (page 13, lines 9-10; Supplementary Fig. 4b). The effect of dicarbonyls often depends on small experimental details and varies between studies, making a comparison of concentration-response relationship with published reports difficult. Therefore, we have tested dimethylglyoxal and methylglyoxal side-by-side and found markedly bigger effects of dimethylglyoxal than methylglyoxal on ROS production (Fig. 8a, b; Supplementary Fig. 4c, d).

In addition, dimethylglyoxal stimulated the expression of pro-inflammatory genes in a concentration of 100 μ M (Fig. 8f, g, *Ccl2*, *Tgfb2*, *Ptges*). In these experiments, we did not observe signs of cell death and the expression of housekeeping genes was unaltered.

Reviewer #2 “5. Neuroinflammation experiments were also performed in DMG-treated mice. Why not perform them in *Ilvbl*-positive mice?”

Response: Please note that *Ilvbl*-positive mice are wild-type littermates of the *Ilvbl*-knockout animals. Experimental diabetes has been shown to lead to neuroinflammatory changes in the brains of wild-type *Ilvbl*^{+/+} mice¹⁸. The reviewer probably suggests to compare neuroinflammatory changes in

Ilvbl^{+/+} wild-type and *Ilvbl*^{-/-} mice. We have quantified the infarct size, brain edema or blood-brain barrier integrity in *Ilvbl*^{-/-} and *Ilvbl*^{+/+} mice but found no difference between the two genotypes. In the revised manuscript we discuss possible explanations for these negative findings (page 16, lines 26-28). As we found still significant dimethylglyoxal concentrations in brain and unchanged levels in blood of *Ilvbl*^{-/-} mice, a likely explanation is that the reduction of dimethylglyoxal production by the *Ilvbl* knockout might not be sufficient to achieve a significant lowering of neuroinflammation.

Reviewer #2 “6. The physiological significance of the experiments is also highlighted by the use of the same dosage for i.p. and oral DMG administration. The authors used 100 mg/kg in both protocols, although they are expected to produce completely different DMG bioavailability. This raises doubts about the physiological meaning of especially the i.p. protocol. Interestingly, in both figures 7 and 9 authors show plasma concentrations in similar ranges (2-5 nM) with such completely different protocols, which is not expected. Authors should address this question and confirm whether this is real or an experimental error.”

Response: We apologize if the description of the dosing regimens was misleading. In the revised manuscript we have stressed the difference (page 12, line 16) and improved the schemes summarizing the dosing regimens (Fig. 7a, Fig. 9b). The dosage for i.p. and oral administration was only nominally the same (100 mg/kg body weight) but followed different time schedules and resulted in different plasma concentrations. We injected a parenteral i.p. bolus of 100 mg/kg body weight to search for dimethylglyoxal adducts and administered the same dose but over one day in the drinking water to investigate functional consequences. Due to the presumably low oral bioavailability and the short half-life of dimethylglyoxal, the two routes of administration led to dramatically different dimethylglyoxal plasma concentrations as expected (Supplementary Fig. 4a, Fig. 9a). The i.p. administration of dimethylglyoxal resulted in peak concentrations above physiological levels. Only 24 h after the i.p. injection, plasma dimethylglyoxal levels were similar to the constant levels achieved by oral administration (36 nM in Fig. 7b, 24 nM in Fig. 9c, 30 nM in Fig. 9d). Importantly, the plasma concentrations obtained with chronic oral administration were similar to those observed in diabetes models.

Reviewer #2 “7. Since authors show the rapid disappearance of DMG from the plasma and suggest a transformation into HBL, what is the evidence about DMG intestinal absorption? Authors show increased DMG plasma levels after 6 weeks of oral administration, but not after 14 weeks. On the other hand, increased HBL levels were shown. Is there evidence of DMG absorption both in the free form and in the HBL form?”

Response: The reviewer raises the key point of whether the reactive compound dimethylglyoxal can be absorbed. To address this important question, we administered ¹³C₄-labeled dimethylglyoxal to mice by gavage. We could unequivocally detect ¹³C₄-labeled dimethylglyoxal in plasma and even in brain tissue of the mice (Fig. 9a). Other labelled dicarbonyls did not occur in plasma or brain suggesting that dimethylglyoxal is not degraded to methylglyoxal or glyoxal. Whether HBL is also resorbed is unclear so far. Based on data for other dicarbonyl adducts, intestinal absorption of HBL seems possible but evaluation of this idea would require synthesis of HBL which has not yet been achieved. Furthermore, we have measured dimethylglyoxal in additional mice after 14 weeks of oral native dimethylglyoxal administration. With the larger sample size, dimethylglyoxal concentrations were now significantly elevated after 14 weeks (Fig. 9d), supporting the notion that orally administered dimethylglyoxal can be absorbed in the intestine.

Reviewer #3 (Remarks to the Author):

“The manuscript by Markus Schwaninger et al. details the discovery of dimethylglyoxal as a neuro-damaging substance derived from pyruvate, which leads to cognitive dysfunction in diabetes. Although the discovery seems serendipitous, the authors meticulously construct a comprehensive series of experiments to trace the origin and destiny of dimethylglyoxal.

The authors have identified α -dicarbonyl dimethylglyoxal as a product of cellular pyruvate metabolism. Although dimethylglyoxal has previously been recognized as a microbial metabolite and flavor compound, its biological activity and origin in mammals remained unclear. Unlike other α -dicarbonyls formed as by-products of glycolysis, dimethylglyoxal was shown to originate from pyruvate metabolism. This process involves the condensation of C2 and C3 atoms from two pyruvate molecules after decarboxylation. Interestingly, the gene *Ilvbl*, homologous to yeast acetolactate synthase, contributes to dimethylglyoxal formation in animals. Elevated dimethylglyoxal levels were detected in the blood and brain tissues of experimental and clinical diabetes cases. It was found to impair blood-brain barrier function and cause cognitive defects in mice similar to those seen in diabetes-induced cognitive impairment. Dimethylglyoxal's synthesis pathway could offer new insights for addressing carbonyl stress. Combining metformin with insulin proved more successful in reducing dimethylglyoxal levels than insulin alone in diabetes patients. This discovery could potentially enhance metformin's efficacy in preventing diabetes complications. Targeting ILVBL, a druggable enzyme, might open up novel treatment avenues for diabetic complications.

The experimental work is nicely developed including an animal model of diabetes, humans, and cell cultures to confront the different hypotheses. The questions that appear while reading the experiment have been approached in the next step and support the hypothesis. In those aspects within my expertise, the materials and methods section has been adequately described. My only comment is that probably the authors could risk a bit more in the title including the name of the compound. Saying that I think the manuscript deserves to be published in *Nature Communications*.”

Response: We appreciate the reviewer's positive evaluation and completely agree that the title should include the compound name. Following this advice, we have reworded it to “The reactive pyruvate metabolite dimethylglyoxal mediates neurological complications in diabetes”.

1. Kruyt ND, Biessels GJ, Devries JH, Roos YB. Hyperglycemia in acute ischemic stroke: pathophysiology and clinical management. *Nat Rev Neurol* **6**, 145-155 (2010).
2. Dinel A-L, André C, Aubert A, Ferreira G, Layé S, Castanon N. Cognitive and Emotional Alterations Are Related to Hippocampal Inflammation in a Mouse Model of Metabolic Syndrome. *PLoS ONE* **6**, e24325 (2011).
3. Li XL, Aou S, Oomura Y, Hori N, Fukunaga K, Hori T. Impairment of long-term potentiation and spatial memory in leptin receptor-deficient rodents. *Neuroscience* **113**, 607-615 (2002).
4. Rom S, *et al.* Hyperglycemia-Driven Neuroinflammation Compromises BBB Leading to Memory Loss in Both Diabetes Mellitus (DM) Type 1 and Type 2 Mouse Models. *Molecular Neurobiology* **56**, 1883-1896 (2019).

5. Sharma AN, Elased KM, Garrett TL, Lucot JB. Neurobehavioral deficits in db/db diabetic mice. *Physiology & Behavior* **101**, 381-388 (2010).
6. Sharma AN, Elased KM, Lucot JB. Rosiglitazone treatment reversed depression- but not psychosis-like behavior of db/db diabetic mice. *J Psychopharmacol* **26**, 724-732 (2012).
7. Crawford SO, *et al.* Association of blood lactate with type 2 diabetes: the Atherosclerosis Risk in Communities Carotid MRI Study. *International Journal of Epidemiology* **39**, 1647-1655 (2010).
8. Wu Y, Dong Y, Atefi M, Liu Y, Elshimali Y, Vadgama JV. Lactate, a Neglected Factor for Diabetes and Cancer Interaction. *Mediators Inflamm* **2016**, 6456018 (2016).
9. Ye W, Zheng Y, Zhang S, Yan L, Cheng H, Wu M. Oxamate Improves Glycemic Control and Insulin Sensitivity via Inhibition of Tissue Lactate Production in db/db Mice. *PLOS ONE* **11**, e0150303 (2016).
10. Zhao L, Dong M, Ren M, Li C, Zheng H, Gao H. Metabolomic Analysis Identifies Lactate as an Important Pathogenic Factor in Diabetes-associated Cognitive Decline Rats. *Mol Cell Proteomics* **17**, 2335-2346 (2018).
11. Kondoh Y, Kawase M, Kawakami Y, Ohmori S. Concentrations of lactate and its related metabolic intermediates in liver, blood, and muscle of diabetic and starved rats. *Research in Experimental Medicine* **192**, 407-414 (1992).
12. Gao H, Jiang Q, Ji H, Ning J, Li C, Zheng H. Type 1 diabetes induces cognitive dysfunction in rats associated with alterations of the gut microbiome and metabolomes in serum and hippocampus. *Biochimica et Biophysica Acta (BBA) - Molecular Basis of Disease* **1865**, 165541 (2019).
13. Fan L, Cacicedo JM, Ido Y. Impaired nicotinamide adenine dinucleotide (NAD⁺) metabolism in diabetes and diabetic tissues: Implications for nicotinamide-related compound treatment. *J Diabetes Investig* **11**, 1403-1419 (2020).
14. Khan MA, *et al.* Hyperglycemia in Stroke Impairs Polarization of Monocytes/Macrophages to a Protective Noninflammatory Cell Type. *J Neurosci* **36**, 9313-9325 (2016).
15. Allaman I, Belanger M, Magistretti PJ. Methylglyoxal, the dark side of glycolysis. *Front Neurosci* **9**, 23 (2015).
16. Hui S, *et al.* Glucose feeds the TCA cycle via circulating lactate. *Nature* **551**, 115-118 (2017).
17. Inderhees J, Schwaninger M. Liver Metabolism in Ischemic Stroke. *Neuroscience*, (2024).

18. Llorián-Salvador M, Cabeza-Fernández S, Gomez-Sanchez JA, de la Fuente AG. Glial cell alterations in diabetes-induced neurodegeneration. *Cell Mol Life Sci* **81**, 47 (2024).

REVIEWERS' COMMENTS

Reviewer #1 (Remarks to the Author):

Thank you to the authors for a very kind, careful and detailed response to my comments. The clarifications, additional references, and more careful wording in the introduction were very helpful. The authors also performed a number of new experiments to address my concerns, including new data from a T2D model and several additional behavioral measures. The authors should be commended for this large amount of additional effort and for their clear improvements to an already very interesting study.

My only remaining concern is that I still am of the opinion that the phrase “neurological complications” overstates the findings described (which are essentially mild deficits in the object place recognition test and no real significant differences in the other behavioral tests; as well as gliosis and occluding levels in mouse brains). While the manuscript is now much more clearly written, I would ask that the authors carefully consider whether this is the most appropriate phrasing for the title.

Minor correction to be made: the order of “3-DG, Gx, MG and DMG” in Fig 1 legend differs slightly from the order in 1g.

Reviewer #2 (Remarks to the Author):

The authors have made a great effort to answer all the comments and questions raised during the first review round. They have included data to answer such questions and significantly improved the quality of the manuscript. I believe it deserves to be published.

Reviewer #3 (Remarks to the Author):

In my opinion the manuscript can be published

Reviewer #1 (Remarks to the Author):

Thank you to the authors for a very kind, careful and detailed response to my comments. The clarifications, additional references, and more careful wording in the introduction were very helpful. The authors also performed a number of new experiments to address my concerns, including new data from a T2D model and several additional behavioral measures. The authors should be commended for this large amount of additional effort and for their clear improvements to an already very interesting study.

My only remaining concern is that I still am of the opinion that the phrase “neurological complications” overstates the findings described (which are essentially mild deficits in the object place recognition test and no real significant differences in the other behavioral tests; as well as gliosis and occluding levels in mouse brains). While the manuscript is now much more clearly written, I would ask that the authors carefully consider whether this is the most appropriate phrasing for the title.

Response: We thank the reviewer for this judicious comment. We agree and have reworded the title to “The reactive pyruvate metabolite dimethylglyoxal mediates neurological consequences of diabetes”. The term consequence is more neutral than complication.

Minor correction to be made: the order of “3-DG, Gx, MG and DMG” in Fig 1 legend differs slightly from the order in 1g.

Response: In Fig. 1, we present the dicarbonyls in the order 3-DG, Gx, MG and DMG.

Reviewer #2 (Remarks to the Author):

The authors have made a great effort to answer all the comments and questions raised during the first review round. They have included data to answer such questions and significantly improved the quality of the manuscript. I believe it deserves to be published.

Response: We thank the reviewer for this feedback.

Reviewer #3 (Remarks to the Author):

In my opinion the manuscript can be published

Response: We thank the reviewer for this feedback.